# Genetic predisposition to uterine leiomyoma is determined by loci for genitourinary development and genome stability

Niko Välimäki[1,2†], Heli Kuisma[1,2†], Annukka Pasanen[3], Oskari Heikinheimo[4], Jari Sjöberg[4], Ralf Bützow[3], Nanna Sarvilinna[1,2,4,5], Hanna-Riikka Heinonen[1,2], Jaana Tolvanen[1,2], Simona Bramante[1,2], Tomas Tanskanen[1,2], Juha Auvinen[6,7], Outi Uimari[8], Amjad Alkodsi[1,2], Rainer Lehtonen[1,2], Eevi Kaasinen[1,2,9], Kimmo Palin[1,2], Lauri A Aaltonen[1,2]*

[1]Department of Medical and Clinical Genetics, University of Helsinki, Helsinki, Finland; [2]Genome-Scale Biology Research Program, Research Programs Unit, University of Helsinki, Helsinki, Finland; [3]Department of Pathology, University of Helsinki and Helsinki University Hospital, Helsinki, Finland; [4]Department of Obstetrics and Gynecology, University of Helsinki and Helsinki University Hospital, Helsinki, Finland; [5]Institute of Biomedicine, Biochemistry and Developmental Biology, University of Helsinki, Helsinki, Finland; [6]Northern Finland Birth Cohorts' Project Center, Faculty of Medicine, University of Oulu, Oulu, Finland; [7]Center for Life Course Health Research, Faculty of Medicine, University of Oulu, Oulu, Finland; [8]Department of Obstetrics and Gynecology, PEDEGO Research Unit, Medical Research Center Oulu, Oulu University Hospital, University of Oulu, Oulu, Finland; [9]Division of Functional Genomics and Systems Biology, Department of Medical Biochemistry and Biophysics, Karolinska Institutet, Stockholm, Sweden

*For correspondence:
lauri.aaltonen@helsinki.fi

†These authors contributed equally to this work

Competing interests: The authors declare that no competing interests exist.

**Abstract** Uterine leiomyomas (ULs) are benign tumors that are a major burden to women's health. A genome-wide association study on 15,453 UL cases and 392,628 controls was performed, followed by replication of the genomic risk in six cohorts. Effects of the risk alleles were evaluated in view of molecular and clinical characteristics. 22 loci displayed a genome-wide significant association. The likely predisposition genes could be grouped to two biological processes. Genes involved in genome stability were represented by *TERT, TERC, OBFC1* - highlighting the role of telomere maintenance - *TP53* and *ATM*. Genes involved in genitourinary development, *WNT4, WT1, SALL1, MED12, ESR1, GREB1, FOXO1, DMRT1* and uterine stem cell marker antigen *CD44*, formed another strong subgroup. The combined risk contributed by the 22 loci was associated with *MED12* mutation-positive tumors. The findings link genes for uterine development and genetic stability to leiomyomagenesis, and in part explain the more frequent occurrence of UL in women of African origin.
DOI: https://doi.org/10.7554/eLife.37110.001

## Introduction

Uterine leiomyomas (ULs), also known as fibroids or myomas, are benign smooth muscle tumors of the uterine wall. They are extremely common; approximately 70% of women develop ULs before menopause (*Stewart et al., 2017*). The symptoms, occurring in one fifth of women, include

**eLife digest** Fibroids – also known as uterine leiomyomas, or myomas – are a very common form of benign tumor that grows in the muscle wall of the uterus. As many as 70% of women develop fibroids in their lifetime. About a fifth of women report symptoms including severe pain, heavy bleeding during periods and complications in pregnancy. In the United States, the cost of treating fibroids is estimated to be $34 billion each year.

Despite the prevalence of fibroids in women, there are few treatments available. Drugs to target them have limited effect and often an invasive procedure such as surgery is needed to remove the tumors. However, a better understanding of the genetics of fibroids could lead to a way to develop better treatment options.

Välimäki, Kuisma et al. used a genome-wide association study to seek out DNA variations that are more common in people with fibroids. Using data from the UK Biobank, the genomes of over 15,000 women with fibroids were analyzed against a control population of over 392,000 individuals. The analysis revealed 22 regions of the genome that were associated with fibroids. These regions included genes that may well contribute to fibroid development, such as the gene *TP53*, which influences the stability of the genome, and *ESR1*, which codes for a receptor for estrogen – a hormone known to play a role in the growth of fibroids. Variation in a set of genes known to control development of the female reproductive organs was also identified in women with fibroids.

The findings are the result of the largest genome-wide association study on fibroids, revealing a set of genes that could influence the development of fibroids. Studying these genes could lead to more effective drug development to treat fibroids. Revealing this group of genes could also help to identify women at high risk of developing fibroids and help to prevent or manage the condition.

DOI: https://doi.org/10.7554/eLife.37110.002

excessive menstrual bleeding, abdominal pain and pregnancy complications (*Stewart et al., 2017*). In most cases, durable treatment options are invasive (*Stewart, 2015*). ULs cause a substantial human and economic burden, and the annual cost of treating these tumors has been approximated to be as high as $34 billion in the United States, higher than the combined cost of treating breast and colon cancer (*Cardozo et al., 2012*).

Earlier studies have indicated strong genetic influence in UL susceptibility based on linkage (*Gross, 2000*), population disparity (*Wise et al., 2012*) and twin studies (*Luoto et al., 2000*). The most striking UL predisposing condition thus far characterized is hereditary leiomyomatosis and renal cell cancer (HLRCC) syndrome, caused by high-penetrance germline mutations in the *Fumarate hydratase* (*FH*) gene (*Multiple Leiomyoma Consortium et al., 2002*; *Launonen et al., 2001*). Genome-wide association studies (GWAS) have proposed several low-penetrance risk loci but few unambiguous predisposing genes have emerged. Cha et al. reported loci in chromosome regions 10q24.33, 11p15.5 and 22q13.1 based on a Japanese patient cohort (*Cha et al., 2011*). The 11p15.5 locus - near the *Bet1 golgi vesicular membrane trafficking protein like* (*BET1L*) gene - was later replicated in Caucasian ancestry (*Edwards et al., 2013a*). The 22q13.1 locus has been replicated in Caucasian, American and Saudi Arabian populations suggesting *trinucleotide repeat containing 6B* (*TNRC6B*) as a possible target gene (*Edwards et al., 2013a*; *Aissani et al., 2015*; *Bondagji et al., 2017*). Further UL predisposition loci have been suggested at 1q42.2 and 2q32.2 by Zhang et al (*Zhang et al., 2015*). and, at 3p21.31, 10p11.21 and 17q25.3 by Eggert et al (*Eggert et al., 2012*). A recent work reported *cytohesin 4* (*CYTH4*) at 22q13.1 as a novel candidate locus in African ancestry (*Hellwege et al., 2017*). While multiple loci and genes have been implicated through these valuable studies it is not straightforward to connect any of them mechanistically to UL development.

Most ULs show somatic site-specific mutations at exons 1 and 2 of the *mediator complex subunit 12* (*MED12*) gene (*Mäkinen et al., 2011*; *Heinonen et al., 2014*). These observations together with further scrutiny of driver mutations, chromosomal aberrations, gene expression, and clinicopathological characteristics have led to identification of at least three mutually exclusive UL subtypes; *MED12* mutant, *Fumarate Hydratase* (*FH*) deficient, as well as *High Mobility Group AT-Hook 2* (*HMGA2*) overexpressing lesions (*Mehine et al., 2016*).

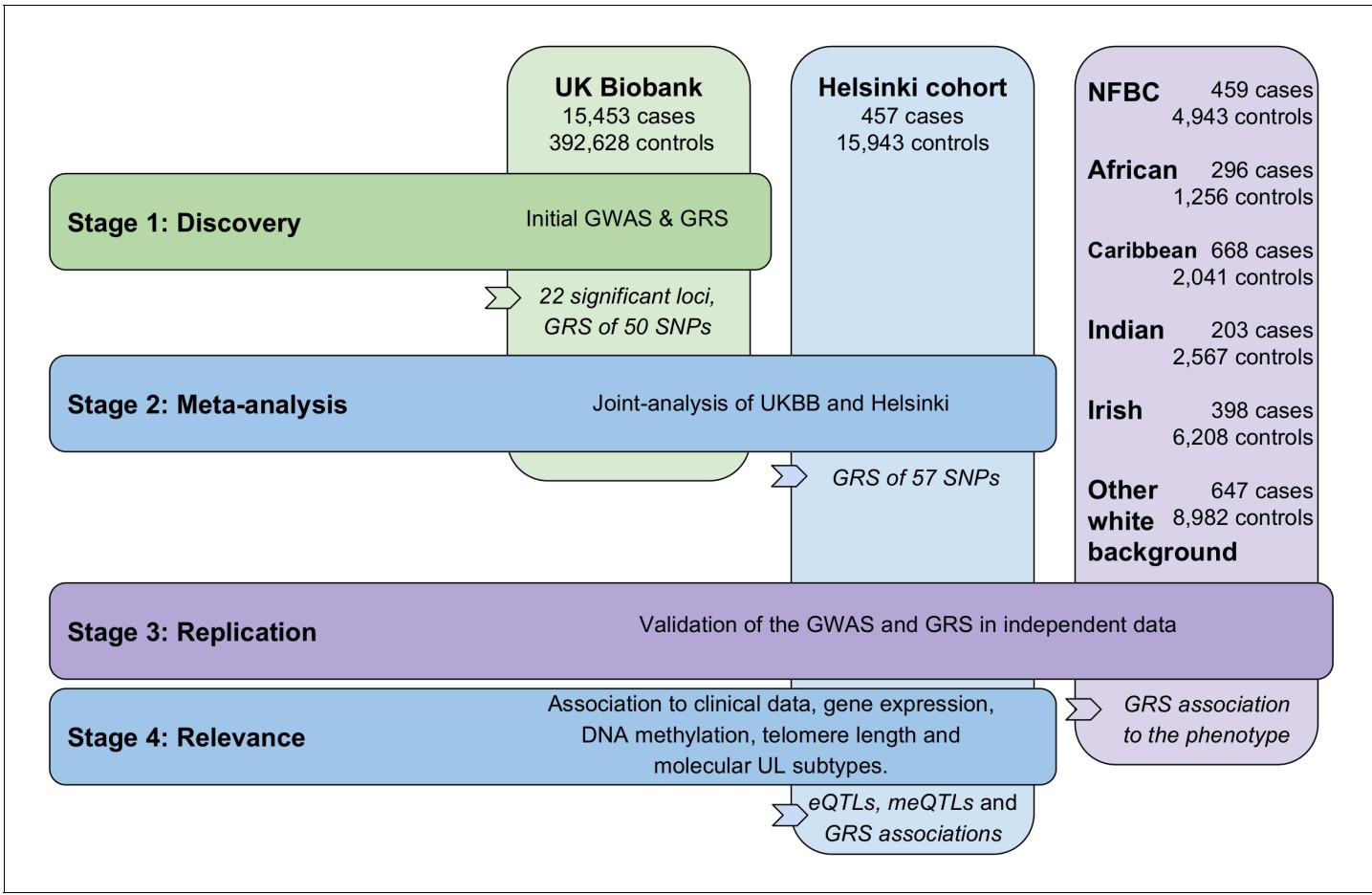

**Figure 1.** Outline of the study stages and genotyping cohorts. GRS, genomic risk score. NFBC, Northern Finland Birth Cohort.
DOI: https://doi.org/10.7554/eLife.37110.003

Here we report the most powerful GWAS on uterine leiomyoma to date, and novel genome-wide significant UL susceptibility loci with plausible adjacent predisposition genes. These genes associate UL genesis to two distinct biological mechanisms: Genome stability related processes are implicated by genes *Tumor Protein P53* (*TP53*) and *ATM Serine/Threonine Kinase* (*ATM*) together with the telomere maintenance genes *Telomerase Reverse Transcriptase* (*TERT*), *Telomerase RNA Component* (*TERC*) and *STN1-CST Complex Subunit* (*OBFC1*). The other prominent group is genes relevant for genitourinary development, specifically *Wnt Family Member 4* (*WNT4*), *Wilms Tumor 1* (*WT1*), *Spalt Like Transcription Factor 1* (*SALL1*), *Estrogen Receptor 1* (*ESR1 or ERα*), *Growth Regulation By Estrogen In Breast Cancer 1* (*GREB1*), *Forkhead Box O1* (*FOXO1*), *Doublesex and Mab-3 Related Transcription Factor 1* (*DMRT1*) and *CD44 Molecule* (*CD44*). Our analysis of the X chromosome identifies a risk allele near *MED12* that drives UL tumorigenesis towards somatic *MED12* mutations. We report altogether 22 genome-wide significant susceptibility loci and compile them into a polygenic risk score. The UL association is then replicated in six independent cohorts of different ethnic origins: individuals of African origin are characterized by the highest risk load. Finally, we investigate the risk alleles' association to clinical features, molecular UL subtypes, telomere length, gene expression and DNA methylation.

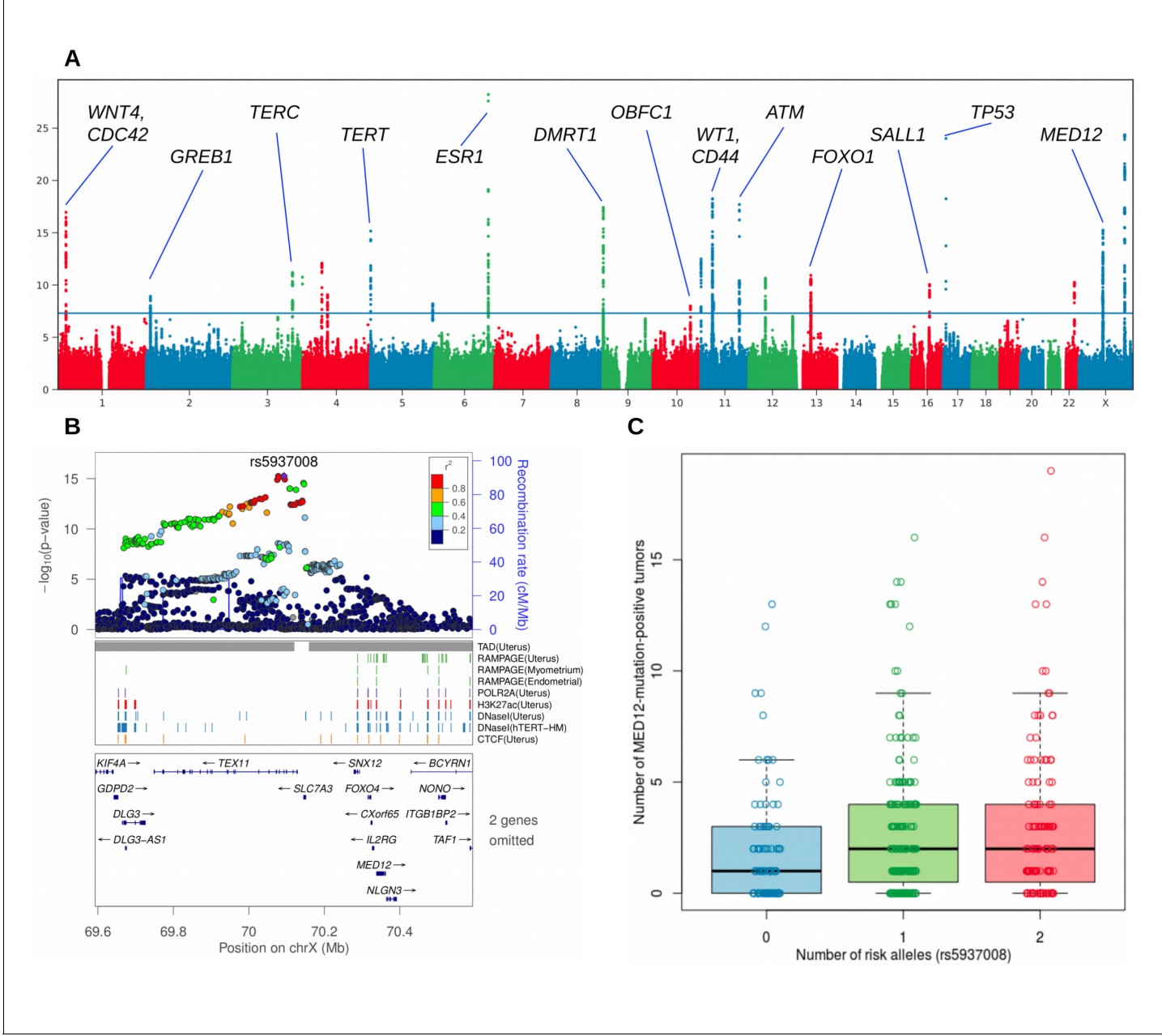

**Figure 2.** Overview of the uterine leiomyoma risk loci and the effect of increased number of MED12-mutated lesions per rs5937008 risk allele. (A), Manhattan plot of the UK Biobank GWAS on 15,453 UL cases and 392,628 controls. On Y-axis, logarithm transformed association values, and on X-axis, autosomes and the X chromosome. The blue horizontal line denotes genome-wide significance ($p=5 \times 10^{-8}$). Gene symbols shown for reference. (B), *MED12* region in more detail. ENCODE tracks (details in Supplementary Methods) are shown for reference. (C), The risk allele near *MED12* (rs5937008) is observed with a significant increase in number of *MED12*-mutation-positive tumors (p=0.009; negative binomial regression; RR = 1.23 per risk allele; n = 457 Helsinki cohort patients).

DOI: https://doi.org/10.7554/eLife.37110.004

## Results

### Identification of predisposition loci

*Figure 1* provides an outline of this study. At discovery stage 1,428 SNPs emerging from 22 distinct genetic loci passed the genome-wide significance level of $5 \times 10^{-8}$. *Figure 2* displays a Manhattan plot of these associations (15,453 UL cases and 392,628 controls; linear mixed model). Two of the

**Table 1.** Predisposition loci for uterine leiomyoma.

| Chr | Position | rs-code | A | B | B freq | OR | P | Likely disease gene |
|---|---|---|---|---|---|---|---|---|
| 6 | 152,562,271 | rs58415480 | C | G | 0.155 | 1.18 | 6.0E-29 | *ESR1* |
| X | 131,312,089 | rs5930554 | T | C | 0.311 | 1.14 | 4.3E-25 | ? |
| 17 | 7,571,752 | rs78378222 [#] | T | G | 0.013 | 1.53 | 9.7E-25 | *TP53* |
| 11 | 32,370,380 | rs10835889 | G | A | 0.159 | 1.14 | 5.5E-19 | *WT1* |
| 11 | 108,149,207 | rs141379009 | T | G | 0.027 | 1.32 | 2.0E-18 | *ATM* |
| 9 | 802,228 | rs7027685 | A | T | 0.402 | 1.11 | 3.8E-18 | *DMRT1* |
| 1 | 22,450,487 | rs2235529 [#] | C | T | 0.157 | 1.14 | 1.1E-17 | *WNT4/CDC42* |
| X | 70,093,038 | rs5937008 | C | T | 0.520 | 0.91 | 5.6E-16 | *MED12* |
| 5 | 1,283,755 | rs72709458 [#] | C | T | 0.206 | 1.12 | 6.9E-16 | *TERT* |
| 11 | 225,196 | rs507139 [*] | G | A | 0.074 | 0.84 | 3.2E-13 | ? |
| 4 | 54,546,192 | rs62323680 | G | A | 0.067 | 1.16 | 8.3E-13 | ? |
| 3 | 169,514,585 | rs10936600 [#] | A | T | 0.244 | 0.91 | 6.4E-12 | TERC |
| 13 | 41,179,798 | rs7986407 | A | G | 0.310 | 1.09 | 1.2E-11 | *FOXO1* |
| 3 | 197,623,337 | rs143835293 | A | G | 0.002 | 1.75 | 1.8E-11 | ? |
| 12 | 46,831,129 | rs12832777 | T | C | 0.701 | 1.09 | 2.3E-11 | ? |
| 22 | 40,669,648 | rs733381 [*] | A | G | 0.213 | 1.10 | 5.7E-11 | ? |
| 16 | 51,481,596 | rs66998222 | G | A | 0.201 | 0.91 | 8.9E-11 | *SALL1* |
| 4 | 70,634,441 | rs2202282 | C | T | 0.497 | 1.07 | 8.7E-10 | ? |
| 2 | 11,702,661 | rs10929757 | A | C | 0.579 | 1.08 | 1.2E-09 | *GREB1* |
| 11 | 35,085,453 | rs2553772 | T | G | 0.538 | 1.07 | 4.4E-09 | *CD44* |
| 5 | 176,450,837 | rs2456181 | C | G | 0.484 | 1.07 | 6.3E-09 | ? |
| 10 | 105,674,854 | rs1265164 | A | G | 0.869 | 0.91 | 1.0E-08 | *OBFC1* |

The numbers for B allele frequency (B Freq), odds-ratio (OR, where B is the effect allele) and association (P) are based on the UKBB cohort (15,453 UL cases). Gene symbols are shown for reference. The genomic coordinates follow hg19 and dbSNP build 147. All genome-wide significant (p<5 × 10$^{-8}$) loci and their highest-association SNP are shown.

[*] Previously implicated predisposition to ULs.

[#] Previous associations to endometriosis, lung adenocarcinoma, glioma or telomere length; see literature in **Appendix 1—table 11**

DOI: https://doi.org/10.7554/eLife.37110.005

significant loci (359/1,428 SNPs) were found on the X chromosome. After linkage disequilibrium (LD; $r^2 \leq 0.3$) pruning the significant SNPs, a total of 50 LD-independent associations remained: the resulting SNPs are given in **Appendix 1—table 2**, and the lead SNPs are summarized in *Table 1*.

*Appendix 1—figure 1* displays the regional structure of each locus together with flanking association values, linkage disequilibrium (LD) and genome annotation. Annotation tracks are included for tissue-specific data on open chromatin, topologically associating domains (TAD) and other regulatory features (details in Supplementary Methods).

## Genomic risk score

A polygenic risk score (*Abraham and Inouye, 2015*) was compiled based on the discovery stage associations. After LD pruning ($r^2 \leq 0.3$) the discovery-stage SNPs, 50 SNPs from the 22 distinct loci passed for the initial genomic risk score (GRS; *Appendix 1—table 4*). The SNP weights were based on UKBB log-odds. We applied this initial GRS of 50 SNPs to the Helsinki cohort and identified a significant association to the UL phenotype (p=8.3 × 10$^{-10}$; adjusted p=1.1 × 10$^{-8}$; one-tailed Wilcoxon rank-sum; W = 1.69 × 10$^6$; 457 cases and 8899 female controls).

## Meta-analysis

The second stage GWAS combined the UKBB and Helsinki cohorts for a meta-analysis approach. The genome-wide statistics revealed rs117245733, at 13q14.11, as the only SNP with a suggestive

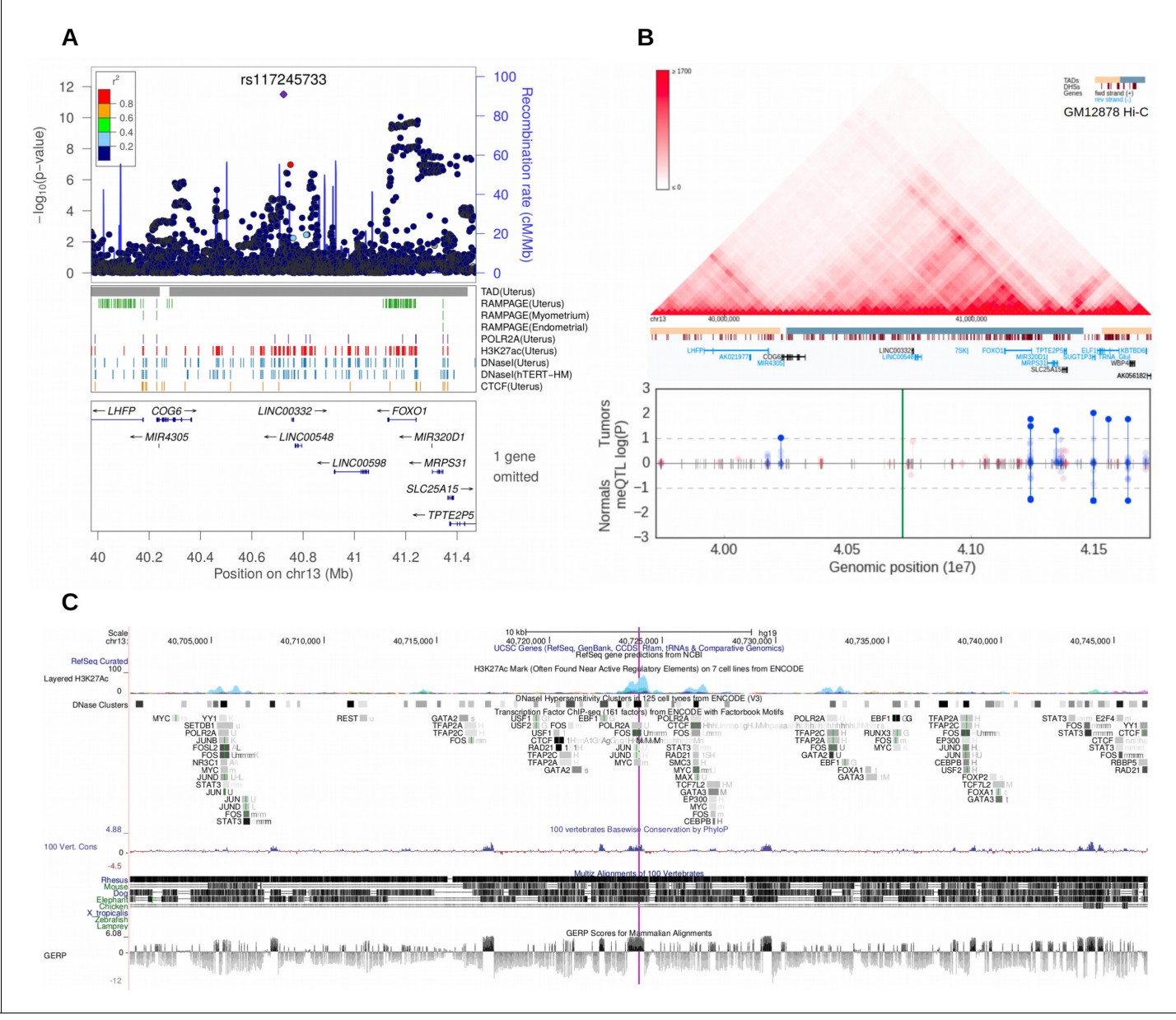

**Figure 3.** Meta-analysis of UL risk revealed rs117245733 at a gene poor region of 13q14.11. (**A**) meta-analysis P-values and the genomic context at the locus. Gene symbols and ENCODE tracks (details in Supplementary Methods) are shown for reference; coordinates follow hg19. (**B**) Hi-C, TADs and CpG methylation around the locus with a 1 Mb flank. The needle plot shows the meQTL associations (dashed lines at 10% FDR; green line denotes the SNP; gray ticks denote all CpGs tested; blue needle for positive coefficient, red for negative coefficient) for tumors (above x-axis; $n_{AA}$ = 53, $n_{AB}$ = 3) and normals (below x-axis; $n_{AA}$ = 33, $n_{AB}$ = 2). (**C**) UCSC genome browser tracks related to conservation and regulation at the locus.
DOI: https://doi.org/10.7554/eLife.37110.006

(p<$10^{-5}$) association in both the UKBB (OR = 1.26; p=4.2 × $10^{-9}$) and Helsinki (OR = 1.82; p=8.1 × $10^{-6}$) cohorts. *Figure 3* shows the regional structure and combined association (fixed effect model p=3.1 × $10^{-12}$) at the locus: the SNP resides on a gene poor region, at a conserved element that displays activity in uterus-specific H3K27ac and DNaseI data (see ENCODE track details in Supplementary Methods). The SNP is independent of the group of associations at *FOXO1* ($r^2$ = 0.0; *Figure 3*).

The meta-analysis identified altogether 112 genome-wide significant SNPs not seen in the discovery stage: seven of those were LD-independent ($r^2$ ≤0.3; *Appendix 1-table 3*) and their UKBB log-

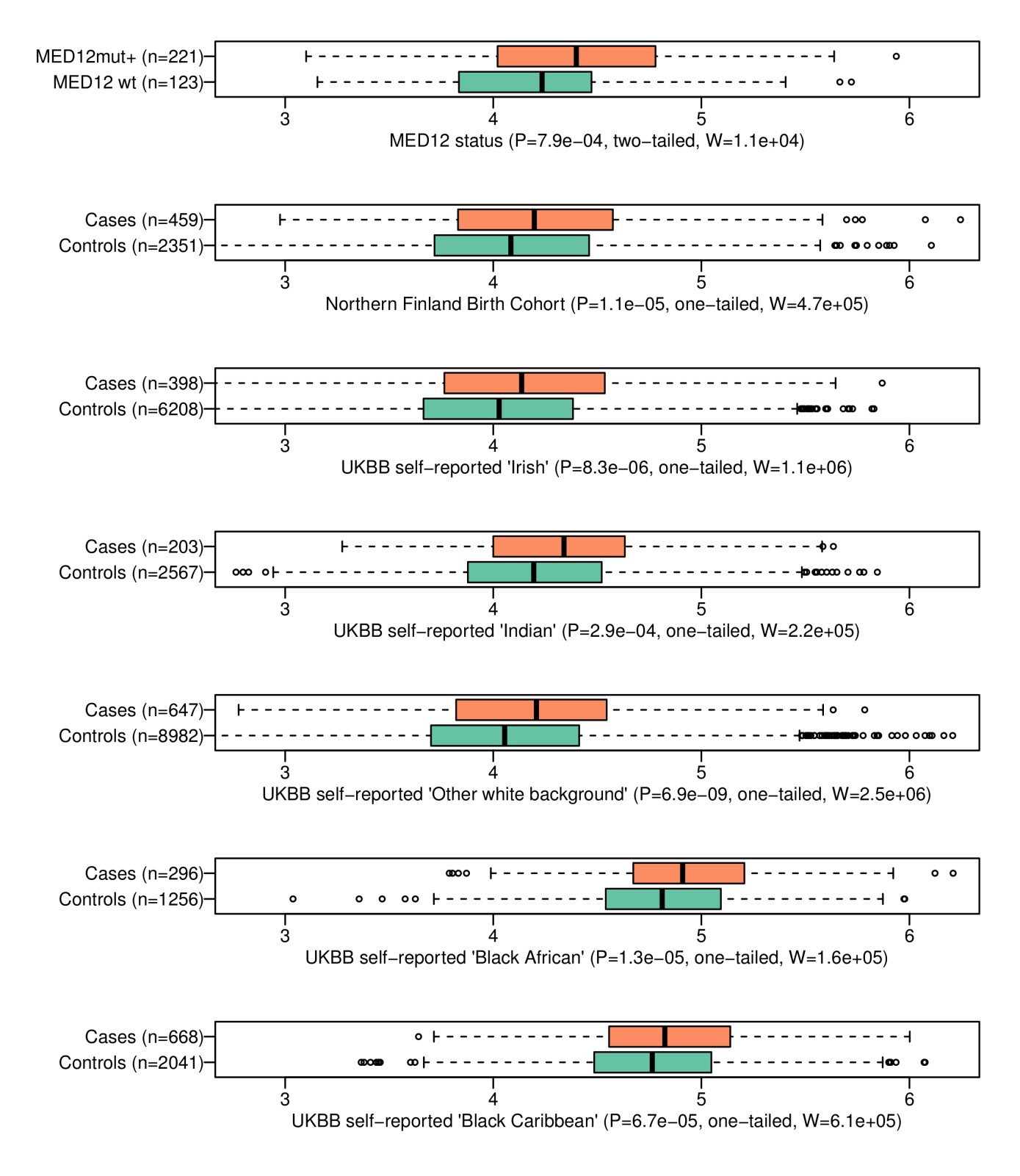

**Figure 4.** The genomic risk score is elevated in patients with MED12-mutated lesions and in respect to the UL phenotype in the six follow-up cohorts. On top, GRS association to *MED12* mutation status. The rest show GRS association to the UL phenotype in six independent replication cohorts. Associations (P) and test statistics (W) are from Wilcoxon rank-sum tests. Only females were included as the control samples. The X-axes show the GRS distributions for each phenotype.

*Figure 4 continued on next page*

*Figure 4 continued*

DOI: https://doi.org/10.7554/eLife.37110.007

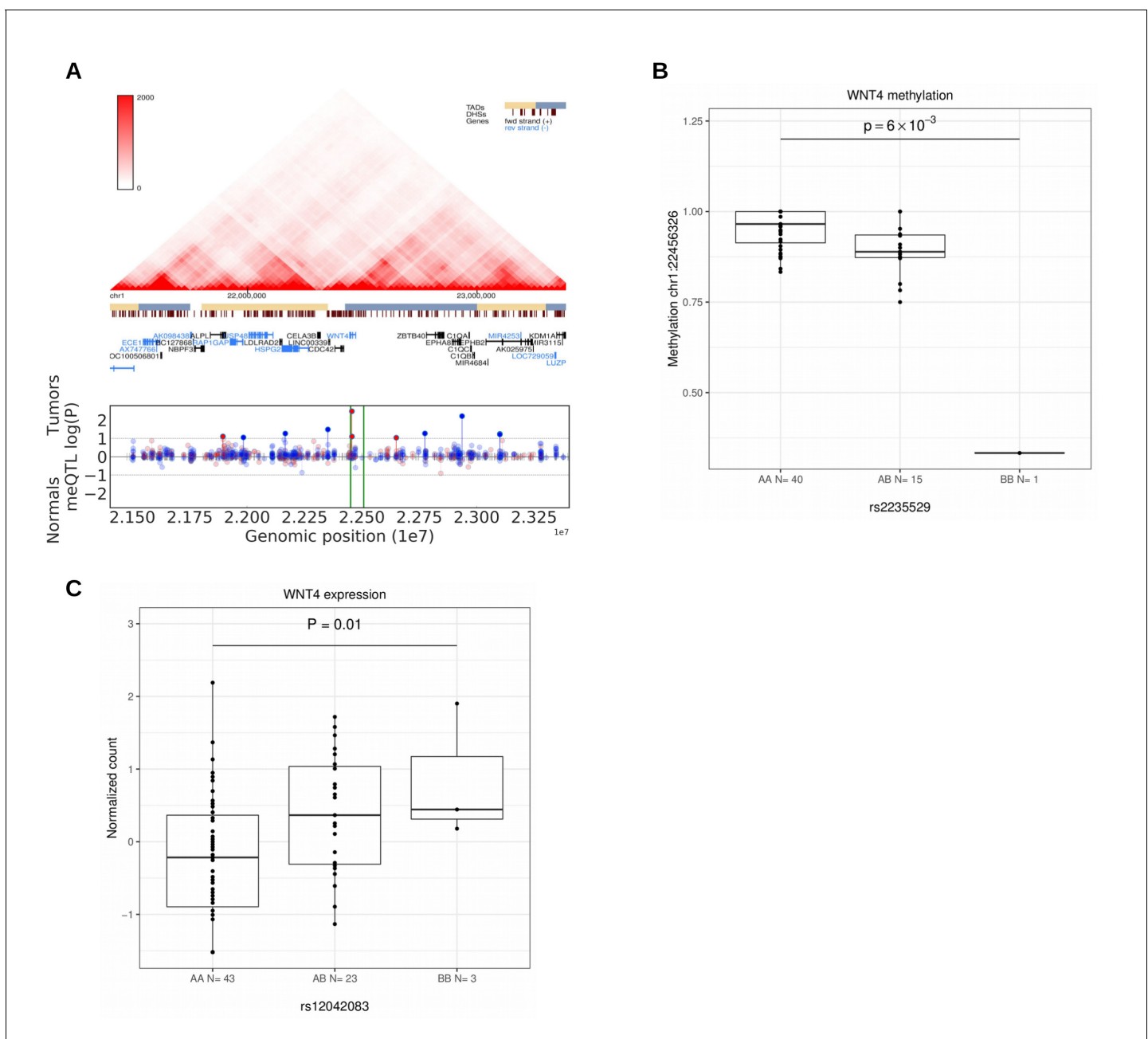

**Figure 5.** Methylation and expression differences in *WNT4*. (**A**), Hi-C, TADs and CpG methylation around the locus with an 1 Mb flank. The needle plot shows the meQTL associations (dashed lines at 10% FDR; green lines denote the two SNPs, rs2235529 and rs2092315; gray ticks denote all CpGs tested; blue needle for positive coefficient, red for negative coefficient) for tumors (above x-axis; $n_{AA}$ = 40, $n_{AB}$ = 15, $n_{BB}$ = 1 for rs2235529 and $n_{AA}$ = 32, $n_{AB}$ = 23 for rs2092315) and normals (below x-axis; $n_{AA}$ = 23, $n_{AB}$ = 12 and $n_{AA}$ = 17, $n_{AB}$ = 17, $n_{BB}$ = 1). (**B**), Methylation differences in tumors (n = 56) at CpG chr1:22456326 by SNP rs2235529. (**C**), *WNT4* expression differences in tumors (n = 41) stratified by the rs12042083 genotype. B is the risk allele, and the P-value is corrected for local multiple testing (permutation based test).

DOI: https://doi.org/10.7554/eLife.37110.008

odds weights were appended to the initial GRS model. The final GRS model of 57 SNPs and their UKBB-based weights is given in *Appendix 1—table 4*. *Supplementary file 1* gives further details on the meta-analysis results and heterogeneity statistics.

## Replication of the GWAS and GRS

The third stage replicated the observations in NFBC and in five different ethnic groups. In NFBC, the SNP identified in the stage two meta-analysis, rs117245733 at 13q14.11, was replicated (p=0.034; linear mixed model; OR = 1.50; 95% CI 1.03 – 2.19). Additional analysis of all 57 SNPs did not reveal other associations: *Supplementary file 2* gives further details on the meta-analysis results and heterogeneity statistics. The association between the GRS and UL phenotype was significant (p=1.1 $\times 10^{-5}$; Wilcoxon rank-sum; adjusted p=1.1 $\times 10^{-4}$; one-tailed; W = 4.7 $\times 10^5$) in NFBC. These case-control distributions of GRS are displayed in *Figure 4*.

UL susceptibility is known to vary by ancestry (*Wise et al., 2012*). Five different ethnic groups - African, Caribbean, Irish, Indian and 'other white' background - were available from the UKBB cohort. A total of 2,212 UL cases and 21,054 female controls could be utilized for replication (*Appendix 1—table 1*). *Supplementary file 3* includes all the 57 SNPs and their summary statistics in these five cohorts, together with heterogeneity estimates. Due to the small cohort sizes, none of the single-SNP associations passed genome-wide significance. The GRS model replicated with a significant phenotype association in all five ethnicities (Appendix 1-Table 6). A summary of test statistics, GRS distributions and the numbers of cases and controls for each population is given in *Figure 4*. A more detailed summary of the GRS model and receiver operating characteristic (ROC) curve of each cohort are given in *Appendix 1—figure 5*.

The self-reported 'Black African' (mean GRS 4.83) had an outstanding risk-load compared to Caucasian (self-reported 'White Irish'; mean GRS 4.04) background (*Figure 4*; Wilcoxon rank-sum p<$10^{-15}$). As expected (*Wise et al., 2012*), the African ethnicity displayed an increased prevalence (19%) compared to the Irish (6%). Assuming that the observed GRS weights have a linear relationship to the true risk, the GRS difference between African and Irish ancestries explains 9.0% of the increased prevalence in the African population.

Similar population-specific GRSs could be estimated for the seven populations in the gnomAD database (*Appendix 1—table 7*). *Appendix 1—figure 6* shows an overview of the GRS for each of the populations. African ancestry has been shown to carry a two-to-three times higher prevalence when compared to Caucasian ancestry (*Wise et al., 2012*). Based on the gnomAD frequencies, the increased GRS of African ancestry explains between 8 – 16% of this population difference.

## Association to clinical variables

The number of ULs per patient had a significant positive association to GRS (negative binomial regression p=0.001; adjusted p=0.0032; rate ratio 1.25; 95% CI 1.09 – 1.43 for one-unit increase in GRS; *Appendix 1—figure 7*). No association was found between GRS and age at hysterectomy (*Appendix 1—table 6*). Testing the 57 GRS SNPs separately did not reveal any associations that pass FDR (*Appendix 1—table 5*).

## Association to *MED12* mutated tumors

Our UL set of 1481 lesions included 1159 (78%) mutation-positive and 322 mutation-negative tumors. The occurrence of mutant tumors did not distribute evenly among the 457 patients. In total 221 (48%) and 123 (27%) patients had all their tumors identified as either *MED12*-mutation-positive or -negative, respectively, suggesting that genetic or environmental factors contribute to the preferred UL type in affected individuals, as previously observed (*Mäkinen et al., 2011*). Indeed, the 221 mutation positive patients were found to have a significantly higher GRS (Wilcoxon rank-sum p=7.9 $\times 10^{-4}$; adjusted p=0.0032; two-sided; W = 1.6 $\times 10^4$). This difference in GRS distributions is visualized in *Figure 4*.

Comparison against the population controls (n = 8899 females) revealed that the above-mentioned patient groups differ by their effect size: the *MED12*-mutation-positive (221) subset of patients had an odds ratio of 2.28 for one-unit increase in GRS (95% CI 1.80 – 2.88) compared to the controls, while the mutation-negative (123) subset had an odds ratio of 1.20 (95% CI 0.88 – 1.66).

Thus, the majority of the compiled case-control association signal had arisen from the *MED12*-mutation-positive subset of the patients.

The number of *MED12*-mutation-positive tumors per patient had a significant positive association to GRS (p=3.2 $\times$ 10$^{-4}$; adjusted p=0.002; negative binomial model rate ratio 1.43; 95% CI 1.13 – 3.83 for one-unit increase in GRS; *Appendix 1—figure 8*). No association between the number of *MED12*-mutation-negative tumors and GRS was found (adjusted p=0.053; *Appendix 1—figure 8*).

The GWAS signal near *MED12* was inspected for any associations to somatic *MED12* mutations. Strikingly, the risk allele (rs5937008) did significantly increase the number of *MED12*-mutation-positive tumors (p=0.0087; negative binomial model rate ratio 1.23; 95% CI 1.05 – 1.44). Among our 457 patients, the median number of *MED12*-mutation-positive tumors increased from one to two for the risk allele carriers. The risk locus and its effect on the number of *MED12*-mutation-positive tumors is visualized in *Figure 2*. An additional analysis of each of the 57 GRS SNPs did not reveal any further associations (*Appendix 1—table 5*).

## Association to gene expression

All the genome-wide significant SNPs from UKBB and the meta-analysis stage (altogether 1,540 SNPs) were tested with a permutation based approach. In total 34 and 24 genes passed the local permutation significance threshold (p<0.05) for tumor and matched myometrium data, respectively (*Appendix 1—table 8*). Among the hits in tumors were *WNT4* (p=0.01; permutation test) and *CDC42* (p=0.03) at 1 p, *TNRC6B* (p=0.02) at 22q, *FOXO1* (p=0.03) at 13q, and *DMRT1* (p=0.04) at 9 p. None of the local associations passed a genome-wide FDR of 10%. No significant association was observed between the risk allele and *MED12* expression (rs5936989; *Appendix 1—figure 11*). The full list of eQTL statistics can be found from *Supplementary file 4*.

## Association to DNA methylation

Our analysis of the 57 GRS SNPs revealed altogether 17,030 (9,466 in tumors and 7564 in matched myometrium) cis methylation quantitative trait loci (cis-meQTL) with nominal p<0.05. Of these, 145 passed a 10% FDR. Of the plausible predisposition genes, *FOXO1*, *TERT* and *WNT4* showed significant meQTL associations (*Appendix 1—table 9*). All the cis-meQTLs and annotation for their genomic context are in *Supplementary file 5*.

## Association to telomere length and structural variants

The UL predisposition loci at *TERT, TERC* and *OBFC1* were examined for an effect on telomere length. Overall the telomere length was significantly shorter in tumors than in adjacent matched myometrium (p=0.01; Kruskal-Wallis), as previously reported (*Rogalla et al., 1995*; *Bonatz et al., 1998*). One of the risk alleles at *TERT* (rs2736100) was significantly associated with shorter telomere length (p=0.01; Kruskal-Wallis) (*Appendix 1—figure 12*). Adjusting for the patient age did not explain away the association. The association was not seen in myometrium. The other two LD-independent SNPs at *TERT*, rs72709458 and rs2853676, or the SNPs at *TERC* (rs10936600) and *OBFC1* (rs1265164) did not show association to telomere length (p=0.24, p=0.57, p=0.07 and p=0.48, respectively; Kruskal-Wallis). The combined effect of *TERT* (rs72709458, rs2736100, rs2853676), *TERC* (rs10936600) and *OBFC1* (rs1265164) had a negative trend with telomere length (p=0.055; linear model 95% CI −408.5 – 4.7 per one risk allele; see *Appendix 1—figure 13*). In whole genome sequencing data, no association was detected between genotype and the number of somatic structural variants.

## Pathway enrichment

The DEPICT framework (*Pers et al., 2015*) was ran using the genome-wide significant SNPs from the UKBB cohort, in total 1,069/1,428 autosomal SNPs. The resulting target gene prioritization, pathway enrichment and tissue enrichment results are given in *Supplementary file 6*. The analysis did not reveal any significant enrichments with the exception of one pathway related to induced stress. *ATM* was the highest ranking target gene, and uterus/myometrium were among the highest ranking tissue types.

## Previously proposed UL predisposition loci

Previous UL association studies (*Cha et al., 2011*; *Zhang et al., 2015*; *Eggert et al., 2012*; *Hellwege et al., 2017*) have reported altogether seven genome-wide significant UL susceptibility loci. Two out of the seven loci - that is, 22q13.1 (at *TNRC6B*) and 11p15.5 (at *BET1L*) - replicated in UKBB using 15,453 cases and 392,628 controls. Cha et al (*Cha et al., 2011*). highlight *OBFC1* (at 10q24.33) as a candidate gene and, while the SNP that they reported does not replicate in UKBB, the *OBFC1* region is identified in our discovery stage (rs1265164; *Table 1*). See *Appendix 1—table 10* for a summary of these results.

## Discussion

The UK Biobank genotype-phenotype data revealed 22 novel predisposition loci for UL, most of them in close proximity to highly plausible predisposition genes. The combined UL risk of these loci was replicated in a subsequent analysis of the polygenic risk score (GRS) in six independent cohorts from different ethnic backgrounds. Our multi-ethnic replication implies that the discovered loci are indeed involved in UL development, and the early UL association studies have likely been underpowered to detect them. Three previously reported loci, at *OBFC1* (*Cha et al., 2011*), *TNRC6B* (*Cha et al., 2011*; *Edwards et al., 2013a*; *Aissani et al., 2015*; *Bondagji et al., 2017*) and *BET1L* (*Cha et al., 2011*; *Edwards et al., 2013b*), were also validated, however, the mechanistic connection to UL development remains obscure for the latter two.

Though simple association is not sufficient to formally prove causality, 14 out of the 22 risk loci harbor plausible predisposition genes. These genes can be divided into two groups: *TERT*, *TERC*, *OBFC1* (all involved in telomere length), *ATM* and *TP53* guard stability of the genome. *ESR1*, *GREB1*, *WT1*, *MED12*, *WNT4*, *FOXO1*, *DMRT1*, *SALL1*, and *CD44* play a role in genitourinary development.

Estrogen is a well-known inducer of UL growth (*Borahay et al., 2015*). The top association at 6q25.2 (rs58415480) resides within intron 107 of *Spectrin Repeat Containing Nuclear Envelope Protein 1* (*SYNE1*), 130 kb downstream of *ESR1*, the latter being the only gene that resides completely within the topologically associating domain (TAD; *Appendix 1—figure 1*). While the role of estrogen in leiomyomagenesis has been firmly established, this is the first genetic evidence to this end. The lead SNP at 2 p resides in the third exon of the gene *GREB1*. *GREB1* is an essential regulatory factor of *ESR1* (*Mohammed et al., 2013*).

*WT1*, *WNT4* and *FOXO1* are central factors in uterine development and in the preparation for pregnancy (decidualization) in endometrium (*Biason-Lauber and Konrad, 2008*; *Hill, 2018*; *Kaya Okur et al., 2016*; *Tamura et al., 2017*). Perturbations in their function are known to have neoplastic potential. The strongest association at 11p13 (rs10835889) is 40 kb downstream of the closest gene *WT1*, at a region with enhancer activity (*Appendix 1—figure 1*). *WT1* is a transcription factor that acts as both a tumor suppressor and an oncogene (*Yang et al., 2007*). The lead SNP at 1p36.12 (rs2235529) resides at the second intron of *WNT4*. The risk allele is associated with suggestive upregulation of *WNT4* (*Figure 5C*). *WNT4* is known to be overexpressed in uterine leiomyomas with *MED12* mutations (*Markowski et al., 2012*), and knock-down of *MED12* in UL cells reduces *WNT4* expression (*Al-Hendy et al., 2017*). The risk locus in 1p36.12 was also associated with several meQTLs suggesting that methylation may have a role in *WNT4* regulation (*Figure 5*). *WNT4* encodes a signaling protein that has a crucial role in sex-determination (*Vainio et al., 1999*), and the WNT signaling pathway has a well-established role in various malignancies such as breast and ovarian cancer (*Peltoketo et al., 2004*). Of note, recent GWAS on gestational duration suggested that binding of the estrogen receptor at *WNT4* is altered by rs3820282 ($r^2$ = 0.92 with our lead SNP) (*Zhang et al., 2017*). Both *WNT4* and *FOXO1* are decidualization markers regulated by *ESR1* (*Kaya Okur et al., 2016*). Though these considerations support *WNT4* as a candidate predisposition gene at this locus, the near-by *CDC42* has been shown to play a role in uterine pathology, in particular endometriosis (*Powell et al., 2016*), and should not be overlooked in further work.

Also *MED12* has been implicated in uterine development in a mouse model (*Wang et al., 2017a*). DMRT1 is a transcription factor associated with male sex-development (*Lindeman et al., 2015*). CD44 is a plausible fibroid stem cell marker (*Mas et al., 2015*). Mutations in *SALL1* and a deletion at the GWAS signal have been associated with Townes-Brocks syndrome, a condition

associated with kidney malformations (*Stevens and May, 2016*). Thus genes involved in genitourinary development are strikingly associated with UL predisposition.

*ATM*, *TP53, TERT, TERC* and *OBFC1* could be involved in uterine neoplasia predisposition through genetic instability and telomere maintenance. The lead SNP at 11q (rs141379009) resides in the 22nd intron of *ATM*, and the SNP at 17 p in the 3'-untranslated region of *TP53*. *ATM* and *TP53* are involved in DNA damage response (*Guleria and Chandna, 2016*), and they are among the relatively few genes that have been found to be recurrently mutated in leiomyosarcoma (*Lee et al., 2017*). *TERT* and *TERC* encode subunits of the telomerase enzyme, which guards chromosomal stability by elongating telomeres (*Blasco, 2005*). In addition *OBFC1* has been associated with telomere maintenance (*Lee et al., 2013*). *TERT* is expressed in germ cells as well as in many types of cancers (*Blasco, 2005*). The neoplasia predisposing effect of the risk alleles at the *TERT* locus (rs72709458; rs2736100; rs2853676) has been overwhelmingly documented (*Appendix 1—table 11*). Previous studies have reported contradicting observations on the effect of rs2736100 on telomere length (*Liu et al., 2014*; *ENGAGE Consortium Telomere Group et al., 2014*; *Lan et al., 2013*; *Melin et al., 2012*; *Choi et al., 2015*). ULs have been shown to display shortened telomeres (*Rogalla et al., 1995*; *Bonatz et al., 1998*), potentially provoking chromosomal instability as the lengths of chromosome telomeres are diminished. In our patient cohort, the risk allele at *TERT* (rs2736100) is significantly associated with shorter telomere length (*Appendix 1—figure 12*), whereas the combined effect of SNPs at *TERT*, *TERC* and *OBFC1* did not reach statistical significance.

GRS associated merely with a susceptibility to the most common UL subtype, *MED12* mutation positive tumors. Indeed it has been known that *MED12*-mutation-positive tumors do not distribute randomly among patients (*Mäkinen et al., 2011*), and our data provide at least a partial explanation to this intriguing finding. An outstanding susceptibility locus was identified 250 kb upstream of *MED12*: our in-house patient cohort - together with a mutation-screening of their 1481 tumors - revealed that the risk allele could facilitate selection of somatic *MED12* mutations. It may be that environmental factors contribute more significantly to genesis of *MED12* wild-type lesions. In our recent study this tumor type was associated with a history of pelvic inflammatory disease, and thus infectious agents could be one underlying factor (*Heinonen et al., 2017*). Obviously, also the power of GWAS to detect genetic associations to rare UL subtypes – such as the *HMGA2* overexpressing or *FH* deficient subtypes – is reduced.

This work highlights several new genetic cornerstones of UL formation, highlights genitourinary development and maintenance of genomic stability as key processes associated with it, and represents another step towards a much-improved understanding of its molecular basis. The proposed risk score can stratify the female population to low and high-risk quartiles that differ by two-fold in their UL risk. The population-specific risk score was inflated towards the African and Caribbean cohorts, which connects the predisposition loci to the excess UL prevalence in these ethnicities. While the increased risk appears minor on an individual level, the population-level burden to women's health arising from these risk loci is highly significant considering the incidence of the condition. Together with the recent progress in molecular tumor characterization and subclassification, the identification of the genetic components of UL predisposition should pave the way towards more sophisticated prevention and management strategies for these extremely common tumors. The risk SNP with the most immediate potential value is that at estrogen receptor alpha, and our findings should fuel much further work on the interplay between individual germline genetics, endogenous and exogenous hormonal exposure, and occurrence and growth rate of UL.

## Materials and methods

### Genome-wide association study

*Figure 1* provides an outline of the four stages that were implemented. The discovery stage was conducted with UK Biobank resources (UKBB; project #32506; accessed on April 10, 2018). The resource included pre-imputed genotypes (version 3; March 2018) for a total of 487,409 samples (486,757 samples for the X chromosome) and 96 million SNPs. The background information on the imputation and data quality control (QC) can be found through the UKBB documentation (www. ukbiobank.ac.uk).

The UL cases were identified on the basis of both the self-reported uterine leiomyoma (UL) phenotype (UKBB data-field 20002: Non-cancer illness code 1351) and International Classification of Diseases (ICD) codes (data-fields 41202 – 41205: Main and secondary diagnosis for ICD10 code D25 and ICD9 code 218). These phenotype data resulted in a total of 20,106 UL cases prior to any sample/genotype QC.

Sample QC was based on the UKBB annotation as follows. In total 409,692 samples passed the initial QC on ethnic grouping (UKBB data-field 22006): self-identified as 'White British', and similar genetic ancestry based on a principal component analysis (PCA) of the genotypes. Further sample QC excluded excess kinship (field 22021; 408,797 samples passed), sex-chromosome aneuploidy (field 22019; 408,241) and inconsistent gender (fields 31 and 22001, and one male with self-reported ULs; 408,081). In total 15,453 UL cases and 392,628 population-matched controls (205,157 females and 187,471 males) passed all these criteria.

Raw genotype calls (UKBB version 2; Affymetrix UK BiLEVE Axiom, or Affymetrix UKBB Axiom array) were available for 805,426 SNPs: after filtering out low genotyping rate (<95%), Hardy-Weinberg equilibrium ($p<10^{-10}$) and minor allele frequency (MAF) <0.001, the remaining 611,887 autosomal genotypes were used to train the mixed model for association testing. Imputed SNPs with MAF <0.001 and imputation score (INFO) <0.3 were excluded. Further SNPs were excluded due to imputation panel differences between cohorts, and the remaining 8.3 million SNPs (Haplotype Reference Consortium, HRC1.1 panel) were tested for case-control association with BoltLMM (version 2.3.2) (*Loh et al., 2015*). The default linear, infinitesimal mixed model was used to adjust for any underlying population structure. The model included categorical covariates for the 22 UK Biobank assessment centres and two genotyping arrays.

## Meta-analysis

The second stage meta-analysis utilized the genome-wide summary statistics from UKBB and the Helsinki cohort of 457 UL cases and 15,943 controls. Details on the Helsinki cohort's imputation, sample and genotype QC are given in the Supplementary Methods. A total of 8.3 million SNPs passed imputation QC and were utilized in the meta-analysis with PLINK (version 1.90b3i) (*Chang et al., 2015*).

## Replication

The SNPs were tested for association in six independent cohorts: Northern Finland Birth Cohort (NFBC) and five non-overlapping subsets of UKBB. In addition to the single-SNP association tests, a polygenic risk score (*Abraham and Inouye, 2015Abraham and Inouye, 2015*) was compiled as follows. The genomic risk score (GRS) was computed as a sum over SNP dosages weighted by their observed log-odds: LD pruning ($r^2 \leq 0.3$) was applied in the order of UKBB association, and the remaining, genome-wide significant SNPs were chosen for the GRS. The log-odds weights were taken from the UKBB statistics (i.e. logarithm of the dosage-based ORs). The resulting GRS model was evaluated using R (3.3.1) and the packages PredictABEL (1.2 – 2) and MASS (7.3 – 45).

The Northern Finland Birth Cohort (NFBC) had in total 459 UL cases and 4943 controls; details of the imputation, sample and genotype QC are given in the Supplementary Methods.

Five non-overlapping, self-reported population-strata were available from UKBB (data-field 21000) and could be utilized as an independent replication: the five self-reported ancestries were 'Black African', 'Black Caribbean', 'Indian', 'White Irish' and 'Other white background'. Sample QC excluded excess kinship (field 22021), sex-chromosome aneuploidy (field 22019) and inconsistent gender (fields 31 and 22001). The numbers of cases and controls that passed the sample QC can be found in *Figure 1*. A summary of background variables is given in *Appendix 1—table 1*. These five sample subsets did not overlap with the discovery GWAS individuals. A collection of ancestry-informative genotypes was utilized to assess the genetic homogeneity of each of the self-reported ancestry (details in Supplementary Methods).

## Patient and tumor material

Our in-house patient and tumor data were investigated regarding the identified risk loci. All tumors of ≥1 cm diameter had been harvested and stored fresh-frozen (details in Supplementary Methods). *MED12* mutations were screened by Sanger sequencing the *MED12* exons 1 and 2 and their flanking

sequences (60 bp) from all uterine leiomyoma and matching normal myometrium samples (*Mäkinen et al., 2011*; *Heinonen et al., 2014*). The resulting sequence graphs were inspected manually and with Mutation Surveyor software (Softgenetics, State College, PA). Clinical patient data was available for the number of ULs, menopause status, parity, body mass index (BMI) and age at hysterectomy (*Appendix 1—figure 2*). This study was conducted in accordance with the Declaration of Helsinki and approved by the Finnish National Supervisory Authority for Welfare and Health, National Institute for Health and Welfare (THL/151/5.05.00/2017), and the Ethics Committee of the Hospital District of Helsinki and Uusimaa (HUS/177/13/03/03/2016).

## Expression quantitative trait loci analysis

For the cis expression quantitative trait loci (cis-eQTL) analysis, genes with less than six reads in over 80% of the samples were filtered out. The between sample normalization was done with Relative Log Expression (RLE) normalization and each gene was inverse normal transformed. The eQTL analysis was run with FastQTL (version 2.184) (*Ongen et al., 2016*) separately for 60 tumors and 56 patient-matched unaffected, adjacent myometrium samples using permutation approach. The permutation parameter was set to '1000 10000'. Sequencing batch was used as a covariate. The cis-region was set to be 2 Mb. FDR correction was applied for tumors and matched myometrium separately.

## Methylation quantitative trait loci analysis

DNA methylation was studied in 56 tumors and 36 matched myometrium samples. The methylation calls were analyzed with bsseq (version 1.12.2) (*Hansen et al., 2012*). Only the methylation in CpG context was considered. Every locus was required to have the coverage of $\geq 2$ in at least 90% of samples. The association between methylation and genotype was studied with MatrixEQTL (version 2.1.1) using a linear regression model (*Shabalin, 2012*). The LD-independent ($r^2 \leq 0.3$) SNPs from the discovery stage (*Appendix 1—table 2*) and meta-analysis (*Appendix 1—table 3*) were considered (altogether 57 SNPs). The SNPs with MAF <0.05 in the methylation samples were filtered out. This resulted in 44 SNPs in tumors and 45 SNPs in matched myometrium. Cis methylation quantitative trait loci (cis-meQTL) was determined to be within 1 Mb flank from the SNP of interest. To annotate the CpGs with genomic context, the overlap between UCSC's gene track (hg19) and known CpG islands was studied. As the role of promoter methylation is well known, promoter methylation was studied in addition to gene body methylation. Core promoter was defined as a region −2 kb and +1 kb from the transcription start site. The methylation analysis was performed separately for tumors and matched normal myometrium to study whether the changes in methylation could be observed in both tissues.

## Whole genome analysis

The whole genome sequenced (WGS) samples, in total 71 tumors (48 Illumina, 23 Complete Genomics) and 51 matched myometrium samples (28 Illumina and 23 Complete Genomics), were prepared following Illumina and Complete Genomics protocols and processed as described previously (*Mehine et al., 2013*). Structural variation was defined as a structural rearrangement (e.g deletion, inversion or translocation) not detectable in matched normal myometrium. Structural variation was detected as described in Mehine et al. (*Mehine et al., 2013*) The mean telomere length was estimated for Illumina samples using Computel (version 0.3) (*Nersisyan and Arakelyan, 2015*) with the default settings. Clonally related tumors were excluded from the analysis by randomly sampling one tumor to represent each clonally related tumor group. Clonally related tumors had identical changes in driver genes and shared at least a subset of somatic copy-number changes and/or copy neutral loss of heterozygosity (see Mehine et al. (*Mehine et al., 2015*) for further details in identification of the clonally related tumors). Kruskal-Wallis test was used to assess the telomere length differences between tumors and matched myometrium as well as between genotypes. Linear model was used to calculate the association between number of risk alleles and telomere length.

## Pathway enrichment

Pathway enrichment of all genome-wide significant SNPs was tested with DEPICT (version 1 release 194) following the default settings (*Pers et al., 2015*). The tool is designed to integrate multiple

GWAS loci for in silico target gene prioritization, pathway enrichment and tissue-specific expression profiling. In short, the DEPICT framework combines phenotype-free co-expression networks, predefined pathways and protein-protein interaction networks in order to reveal functionally connected genes among the multiple risk loci. The tool is restricted to autosomal SNPs.

## Statistical analysis

Meta-analysis was implemented with an inverse-variance weighted, fixed effect model. Associations between the risk alleles and other variables were tested assuming an additive genotype model unless otherwise noted. The DHARMa (0.1.5) package was applied to evaluate the goodness-of-fit of the binomial and negative binomial models. The contribution of GRS to prevalence was estimated by $[(E_a/P_i-1)/(P_a/P_i-1)]$, where $E_a = P_i*GRS_a/GRS_i$ assumes a linear relationship between GRS and the true risk, and $P_x$ and $GRS_x$ are the population-specific prevalence and mean GRS, respectively. All statistical tests were two-tailed unless otherwise noted.

Summary statistics were collected from each of the study stages and are available as *Appendix 1—table 2* and *Supplementary file 1–3*. For each SNP, we report its allele frequency, effect size estimates and association based on the default linear, infinitesimal mixed model. For the meta-analysis stages, we also report the Cochrane's Q statistic and $I^2$ heterogeneity index in addition to the fixed-effects meta-analysis association and effect size (random-effects meta-analysis is included for reference). BoltLMM reported lambda ($\lambda$GC) 1.055, 1.045 and 1.016 for UKBB, Helsinki and NFBC, respectively. The X chromosome associations were processed separately and included only the female controls ($\lambda$GC 1.052, 1.049 and 1.005 for UKBB, Helsinki and NFBC, respectively).

For GWAS, $p < 5 \times 10^{-8}$ was reported as significant. The GRS association tests (*Appendix 1—table 6*) were controlled for family-wise error rate (FWER) and reported significant for Holm-Bonferroni adjusted $p < 0.05$. Large families of association tests were controlled for false discovery rate (FDR; Benjamini-Hochberg method) and noted significant at FDR < 10%. In the six telomere length association tests and the two structural variation association tests, $p < 0.05$ was considered statistically significant.

## Acknowledgements

We are thankful to Sini Marttinen, Sirpa Soisalo, Marjo Rajalaakso, Inga-Lill Åberg, Iina Vuoristo, Alison Ollikainen, Elina Pörsti, Salla Välipakka and Heikki Metsola for their technical support. We also thank Pirjo Ikonen and the rest of the staff of the Kätilöopisto Maternity Hospital, and the staff of the Department of Pathology, University of Helsinki for technical assistance. We thank Minna Männikkö, Tuula Ylitalo, and the rest of the staff of the Northern Finland Birth Cohort Studies, Faculty of Medicine, University of Oulu for technical assistance. The study was supported by grants from Academy of Finland (Finnish Center of Excellence Program 2012 – 2017, 2018 – 2025, No. 1250345 and 312041), European Research Council (ERC, 695727), Cancer Society of Finland, Sigrid Juselius Foundation and Jane and Aatos Erkko Foundation. NV received a grant from the Academy of Finland (No. 287665). KP received a grant from Nordic Information for Action eScience Center (NIASC), the Nordic Center of Excellence financed by NordForsk (No. 62721). This research has been conducted using the UK Biobank Resource, project #32506.

## Additional information

### Funding

| Funder | Grant reference number | Author |
|--------|------------------------|--------|
| Terveyden Tutkimuksen Toimikunta | 1250345 | Lauri A Aaltonen |
| European Research Council | 695727 | Lauri A Aaltonen |
| Syöpäjärjestöt | | Lauri A Aaltonen |
| Sigrid Juséliuksen Säätiö | | Lauri A Aaltonen |
| Jane ja Aatos Erkon Säätiö | | Lauri A Aaltonen |

| Luonnontieteiden ja Tekniikan Tutkimuksen Toimikunta | 287665 | Niko Välimäki Niko Välimäki |
| NordForsk | 62721 | Kimmo Palin |
| Terveyden Tutkimuksen Toimikunta | 312041 | Lauri A Aaltonen |

The funders had no role in study design, data collection and interpretation, or the decision to submit the work for publication.

## Author contributions

Niko Välimäki, Heli Kuisma, Formal analysis, Writing—original draft; Annukka Pasanen, Oskari Heikinheimo, Jari Sjöberg, Ralf Bützow, Nanna Sarvilinna, Juha Auvinen, Outi Uimari, Resources; Hanna-Riikka Heinonen, Jaana Tolvanen, Simona Bramante, Tomas Tanskanen, Amjad Alkodsi, Rainer Lehtonen, Eevi Kaasinen, Formal analysis; Kimmo Palin, Lauri A Aaltonen, Supervision

## Author ORCIDs

Niko Välimäki http://orcid.org/0000-0001-9200-9560
Annukka Pasanen http://orcid.org/0000-0002-0079-9807
Jaana Tolvanen http://orcid.org/0000-0003-1183-4943
Amjad Alkodsi http://orcid.org/0000-0003-3528-4683
Kimmo Palin http://orcid.org/0000-0002-4621-6128
Lauri A Aaltonen http://orcid.org/0000-0001-6839-4286

## Ethics

Human subjects: The anonymous patient samples (65) were collected according to Finnish laws and regulations by permission of the director of the health care unit. For the rest of the patients, an informed consent was obtained. This study was conducted in accordance with the Declaration of Helsinki and approved by the Finnish National Supervisory Authority for Welfare and Health, National Institute for Health and Welfare (THL/151/5.05.00/2017), and the Ethics Committee of the Hospital District of Helsinki and Uusimaa (HUS/177/13/03/03/2016).

## Decision letter and Author response

Decision letter https://doi.org/10.7554/eLife.37110.046
Author response https://doi.org/10.7554/eLife.37110.047

# Additional files

## Supplementary files

• Supplementary file 1. Summary statistics for the UKBB and Helsinki cohorts. For each of the 57 GRS SNPs, we report the allele frequency, association effect size and P-value, together with the heterogeneity estimates Cochrane's Q-value and $I^2$ index.
DOI: https://doi.org/10.7554/eLife.37110.009

• Supplementary file 2. Summary statistics for UKBB, Helsinki and NFBC. For each of the 57 GRS SNPs, we report the allele frequency, association effect size and P-value, together with the heterogeneity estimates Cochrane's Q-value and $I^2$ index.
DOI: https://doi.org/10.7554/eLife.37110.010

• Supplementary file 3. Summary statistics for the five UKBB follow-up cohorts. For each of the 57 GRS SNPs, we report the allele frequency, association effect size and P-value, together with the heterogeneity estimates Cochrane's Q-value and $I^2$ index.
DOI: https://doi.org/10.7554/eLife.37110.011

• Supplementary file 4. All the cis-eQTL summary statistics. Tumors (T) and myometrium normal tissues (N) were analyzed separately. For each SNP, we report the local permutation test results from FastQTL (details in the Methods section).
DOI: https://doi.org/10.7554/eLife.37110.012

• Supplementary file 5. All the cis-meQTL summary statistics and annotation for their genomic context. Tumors and myometrium normal tissues were analyzed separately. The association statistics are based on MatrixEQTL (details in the Methods section).
DOI: https://doi.org/10.7554/eLife.37110.013

• Supplementary file 6. Pathway-based analysis of the genome-wide significant SNPs. Includes target gene prioritization, pathway enrichment and tissue-specific expression profiling results from the DEPICT framework (details in the Methods section).
DOI: https://doi.org/10.7554/eLife.37110.014

• Transparent reporting form
DOI: https://doi.org/10.7554/eLife.37110.015

### Data availability

The UKBB data is available through the UK Biobank (http://www.ukbiobank.ac.uk). The NFBC data can be requested from the Northern Finland Birth Cohorts' Project Center at the Medical Faculty, University of Oulu (http://www.oulu.fi/nfbc/). The summary statistics that support the findings presented in this work are included in Supplementary Tables and Supplementary Data.

The following previously published datasets were used:

| Author(s) | Year | Dataset title | Dataset URL | Database, license, and accessibility information |
|---|---|---|---|---|
| Clare Bycroft, Colin Freeman, Desislava Petkova, Gavin Band, Lloyd T. Elliott, Kevin Sharp, Allan Motyer, Damjan Vukcevic, Olivier Delaneau, Jared O'Connell, Adrian Cortes, Samantha Welsh, Alan Young, Mark Effingham, Gil McVean, Stephen Leslie, Naomi Allen, Peter Donnelly, Jonathan Marchini | 2018 | UK Biobank | https://www.ebi.ac.uk/ega/studies/EGAS00001002399 | Publicly available at the European Genome-phenome Archive (accession no. EGAS00001002399) |
| Leena Peltonen, Aarno Palotie, Nelson Freimer, Joel Hirschhorn, Mark Daly, Chiara Sabatti, Marjo-Riitta Järvelin, Paul Elliott, Mark McCarthy, Stacey Gabriel | 2018 | Northern Finland Birth Cohort | https://www.ncbi.nlm.nih.gov/projects/gap/cgi-bin/study.cgi?study_id=phs000276.v2.p1 | Publicly available at the NCBI dbGaP website (accession no. phs000276.v2.p1) |

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

# Appendix 1

DOI: https://doi.org/10.7554/eLife.37110.016

## Supplementary methods

### UK Biobank

The UK Biobank (UKBB) individuals were divided into six distinct subsets based on their self-reported ancestry. A summary of the background variables for each of the six ancestries is given in *Appendix 1—table 1*.

Sample QC for the discovery subset was readily available from the UKBB annotation (data-field 22006): self-identified as 'White British', and similar genetic ancestry based on a principal components analysis (PCA) of the genotypes.

The small replication cohorts were assessed for genetic homogeneity based on PCA of ancestry informative markers as follows. We pooled together 21 panels of ancestry informative markers - in total 1396 unique, autosomal SNPs, see Soundararajan et al (*Soundararajan et al., 2016*). for references - and compared the resulting PCs against the self-reported ancestry information. As expected, the genetic differences at these informative markers separated each self-reported ancestry into a distinct, homogeneous cluster. See *Appendix 1—figure 4* for the resulting clustering of the follow-up cohorts.

### Helsinki cohort

Our patient cohort (Helsinki cohort) was collected in accordance with the Declaration of Helsinki and approved by the Finnish National Supervisory Authority for Welfare and Health, National Institute for Health and Welfare (THL/151/5.05.00/2017), and the Ethics Committee of the Hospital District of Helsinki and Uusimaa (HUS/177/13/03/03/2016). In total 1577 uterine leiomyoma and corresponding normal myometrial samples were collected as fresh-frozen from 480 patients undergoing hysterectomy as previously described (*Mäkinen et al., 2011*; *Heinonen et al., 2014*; *Heinonen et al., 2017*). See below details on numbers of samples that passed imputation QC.

The number of ULs per patient was determined by the number of ULs harvested from the hysterectomy specimens. The study materials were derived from six different tissue collections, one consisting of anonymous patients and the other five collections consisting of patients who signed an informed consent before entering the study. Pathologists dissected the hysterectomy specimens and collected all visible tumours from each patient, with the exception of one tissue collection in which all feasible distinct tumours $\geq$ 1 cm in diameter were harvested. The smallest lesion used in this study had a diameter of 4 mm. All the specimens underwent routine diagnostic pathological scrutiny, and the histopathological diagnosis for the study samples was retrieved from the pathology reports.

Population-matched control data were obtained from the National FINRISK Study containing 16,048 genotyped individuals (https://www.thl.fi/fi/web/thlfi-en/research-and-expertwork/population studies/the-national-finrisk-study).

### Northern Finland cohort

The Northern Finland Birth Cohort (NFBC) is a prospective collection of Oulu and Lapland region individuals born in 1966. For further validation of our results, the NFBC Project Center (University of Oulu) provided phenotype information on both clinical ICD disease records and 46 year follow-up questionnaires. The genotyped individuals (dbGaP Study Accession: phs000276.v2.p1) had in total 459 UL cases and 4943 population-matched controls.

## Genotyping arrays

The Helsinki cohort was genotyped with Illumina Infinium HumanCore-24 BeadChip. The control samples in the Helsinki cohort were genotyped with the Illumina HumanCoreExome SNP array. The NFBC individuals were genotyped with the Illumina Infinium SNP array. All genomic coordinates follow GRCh37 and dbSNP build 147.

## SNP array data processing and imputation

The Helsinki cohort B-allele frequencies and log-R ratios were extracted with Illumina GenomeStudio software, and somatic allelic imbalance (AI) regions were calculated for all tumors using BAFsegmentation (*Kim et al., 2015*) with default parameters.

Quality control (QC) was implemented using PLINK (v1.90b3i; http://www.cog-genomics. org/plink/1.9/). The Helsinki cohort was inspected for outliers, close relatedness and low genotyping rate: 457 UL cases (representing 1481 tumors collected) and 15,943 controls passed the initial genotyping control. Further genotype QC was implemented to exclude SNPs with low genotyping rate (<95%), excess homozygosity (i.e. homozygotes that exceed respective heterozygotes), rare homozygotes (minor allele frequency, MAF <0.02), Hardy-Weinberg equilibrium ($p<10^{-3}$) or incorrect strand assignment based on LD. The remaining 211,967 SNPs were imputed using the Haplotype Reference Consortium panel (HRC; release 1.1) at the Sanger Imputation service (EAGLE2 +PBWT; https://imputation.sanger.ac.uk). All the reported alleles follow the strand orientation in HRC1.1 and GRCh37 coordinates. The same quality controls were applied to the NFBC data: all 5402 samples passed QC, and the GRS SNPs were imputed following the same process as for the Helsinki cohort. Post-imputation QC of Helsinki and NFBC data excluded SNPs with MAF <0.005 and imputation score (INFO) <0.4. A total of 8.3 million SNPs passed imputation quality.

## Clinical background information

Clinical data were collected based on a retrospective review of medical records of the Helsinki study subjects. Details are provided in Heinonen et al (*Heinonen et al., 2017*). Information on parity, BMI, number of leiomyomas and age at hysterectomy were quantified for a total of n = 367, 366, 457 and 392 patients, respectively. Menopausal status was recorded for 367 patients as either *pre*, *post* or current *HRT* (hormone replacement therapy). *Appendix 1—figure 2* shows the summary statistics of each background variable.

## RNA sequencing

RNA-seq libraries were prepared according to the standard quality requirements for Illumina TruSeq Stranded total-RNA (RiboZero) kit. Paired-end Illumina (HiSeq2500) sequencing produced around 60 to 70 million 2 × 125 bp reads per sample. Data were aligned to human reference transcript (GRCh37) with HISAT2 (2.1.0) using parameter -dta and setting rna-strandness to RF (*Kim et al., 2015*). The alignments were assembled with StringTie (1.3.3b) using parameters -e and -rf (*Pertea et al., 2015*). Data quality was controlled by checking HISAT2 mapping statistics, ribosomal RNA contamination (based on Ensembl annotation release 75) and batch effects (principal components analysis).

## DNA methylation

The SureSelect target enrichment system (Agilent Technologies, Inc., CA, USA) covering 84.5 Mb of the genome was used to prepare bisulfite-sequencing samples. Sample preparations were done according to the manufacturer's instructions. Illumina paired-end sequencing of 56 tumor and 36 normal samples was done in Karolinska Institutet (Sweden) using 100 bp read length and the HiSeq2000 platform.

Raw sequencing reads were quality and adapter trimmed with cutadapt version 1.3 in Trim Galore. Low-quality ends trimming was done using Phred score cutoff 30. Adapter trimming

was performed using the first 13 bp of the standard Illumina paired-end adapters with stringency overlap two and error rate 0.1. Read alignment was done against hg19/GRCh37 reference genome downloaded from UCSC Genome Browser with Bismark (version v0.10.0) (*Krueger and Andrews, 2011*) and Bowtie 2 (version 2.0.0-beta6) (*Langmead and Salzberg, 2012*). Duplicates were removed using the Bismark deduplicate function. Extraction of methylation calls was done with Bismark methylation extractor discarding the first 10 bp of both reads and reading methylation calls of overlapping parts of the paired reads from the first read (–no_overlap parameter).

## LocusZoom visualization and ENCODE tracks

LocusZoom (http://locuszoom.org/) plots include the ENCODE tracks for DNase I hypersensitive sites for hTERT-HM (ENCFF001SPI, ENCFF001UXF) and uterus (ENCFF689EGI). RAMPAGE tracks were included for uterus (ENCFF979EGO), myometrium (ENCFF605TKQ) and endometrial microvascular endothelial cells (ENCFF440YZN) experiments. Topologically associating domains from endometrial microvascular endothelial cells (HiC; ENCFF633ORE) and H3K27ac ChIP-seq from uterus (stable peaks; ENCFF045LNJ) were included. Additional uterus-specific ChIP-seq data for *CTCF* (ENCFF282BOE, ENCFF634DDY) and *POLR2A* (ENCFF822OTY, ENCFF164YIY) were also included. Hi-C figures were produced with the '3D genome browser' (http://promoter.bx.psu.edu/hi-c/; GM12878) (*Wang et al., 2017b*).

**Appendix 1—table 1.** Summary of the UKBB cohort individuals.

The UKBB cohort was split into six disjoint, self-reported ancestries. For each ancestry, we summarized the background information for the phenotype (number of UL cases and female controls), proportion of cases (%), age at first assessment visit (mean and SD), number of live births (mean and SD), body mass index (BMI; mean and SD), proportion of hysterectomy cases (%) and age at hysterectomy (mean and SD).

| Self-reported ancestry | Phenotype | N | Cases (%) | Age at first assessment visit | | Number of live births | | BMI | | Age at hysterectomy | | |
|---|---|---|---|---|---|---|---|---|---|---|---|---|
| | | | | Mean | SD | Mean | SD | Mean | SD | Ever had hyst. (%) | Mean | SD |
| White British | cases | 15453 | 7.0 | 56.9 | 7.5 | 1.7 | 1.2 | 27.7 | 5.2 | 31.9 | 46.1 | 9.0 |
| | controls | 205157 | | 56.7 | 7.9 | 1.8 | 1.2 | 27.0 | 5.1 | 7.0 | 42.2 | 10.3 |
| Black African | cases | 296 | 19.1 | 49.6 | 6.8 | 2.0 | 1.7 | 31.2 | 5.6 | 22.1 | 45.1 | 8.8 |
| | controls | 1256 | | 51.8 | 8.1 | 2.8 | 1.8 | 31.3 | 5.7 | 6.5 | 36.4 | 15.4 |
| Black Caribbean | cases | 668 | 24.7 | 50.6 | 6.8 | 1.5 | 1.4 | 30.0 | 6.3 | 24.0 | 41.2 | 12.1 |
| | controls | 2041 | | 53.0 | 8.2 | 2.2 | 1.7 | 29.8 | 5.9 | 9.7 | 37.2 | 14.8 |
| White Irish | cases | 398 | 6.0 | 56.5 | 7.6 | 1.8 | 1.4 | 27.8 | 5.2 | 32.6 | 46.6 | 7.2 |
| | controls | 6208 | | 56.4 | 8.1 | 1.9 | 1.4 | 26.9 | 5.0 | 7.5 | 42.9 | 10.8 |
| Asian Indian | cases | 203 | 7.3 | 53.8 | 7.9 | 2.0 | 1.1 | 27.6 | 4.2 | 33.5 | 44.1 | 11.9 |
| | controls | 2567 | | 53.7 | 8.0 | 2.0 | 1.2 | 27.1 | 4.8 | 6.6 | 39.8 | 13.9 |
| Other white background | cases | 647 | 6.7 | 54.8 | 7.9 | 1.3 | 1.2 | 26.7 | 5.3 | 23.6 | 46.5 | 8.6 |
| | controls | 8982 | | 54.6 | 8.3 | 1.6 | 1.2 | 26.4 | 5.2 | 5.4 | 42.1 | 11.0 |

DOI: https://doi.org/10.7554/eLife.37110.017

**Appendix 1—table 2.** Discovery stage GWAS for the UKBB cohort. The information for B allele frequency (B Freq), odds-ratio (OR) and association (Beta; standard error of Beta; $\chi^2$ and P) were collected from the UKBB cohort. All genome-wide significant, LD-independent ($r^2 \leq 0.3$ pruned in order of UKBB association) SNPs that passed imputation QC are shown. SNPs with $r^2$ = NA are the lead-SNPs of each distinct locus. The A and B alleles are on GRCh37 forward strand. OR was computed from SNP dosages and B as the effect allele. Rows are sorted by genomic position.

| rs-code | Chr | Position | A | B | $r^2$ (reference SNP) | B freq | OR | Beta | SE | $\chi^2$ | P |
|---|---|---|---|---|---|---|---|---|---|---|---|
| rs2235529 | 1 | 22450487 | C | T | NA | 0.157 | 1.14 | −0.005 | 5.81E-04 | 73.373 | 1.1E-17 |
| rs2092315 | 1 | 22507684 | C | T | 0.14 (rs2235529) | 0.248 | 1.07 | −0.003 | 4.90E-04 | 30.465 | 3.4E-08 |
| rs10929757 | 2 | 11702661 | A | C | NA | 0.579 | 1.08 | −0.003 | 4.32E-04 | 36.947 | 1.2E-09 |
| rs11674184 | 2 | 11721535 | T | G | 0.30 (rs10929757) | 0.386 | 0.94 | 0.002 | 4.36E-04 | 30.082 | 4.1E-08 |
| rs10936600 | 3 | 169514585 | A | T | NA | 0.244 | 0.91 | 0.003 | 4.91E-04 | 47.191 | 6.4E-12 |
| rs143835293 | 3 | 197623337 | A | G | NA | 0.002 | 1.75 | −0.043 | 6.33E-03 | 45.143 | 1.8E-11 |
| rs62323680 | 4 | 54546192 | G | A | NA | 0.067 | 1.16 | −0.006 | 8.49E-04 | 51.212 | 8.3E-13 |
| rs2202282 | 4 | 70634441 | C | T | NA | 0.497 | 1.07 | −0.003 | 4.22E-04 | 37.593 | 8.7E-10 |
| rs72709458 | 5 | 1283755 | C | T | NA | 0.206 | 1.12 | −0.004 | 5.28E-04 | 65.165 | 6.9E-16 |
| rs2736100 | 5 | 1286516 | C | A | 0.23 (rs72709458) | 0.498 | 0.91 | 0.003 | 4.23E-04 | 60.879 | 6.1E-15 |
| rs2853676 | 5 | 1288547 | T | C | 0.23 (rs2736100) | 0.731 | 0.91 | 0.003 | 4.77E-04 | 49.761 | 1.7E-12 |
| rs2456181 | 5 | 176450837 | C | G | NA | 0.484 | 1.07 | −0.002 | 4.24E-04 | 33.743 | 6.3E-09 |
| rs4870084 | 6 | 152543949 | C | T | 0.29 (rs6904757) | 0.189 | 0.92 | 0.003 | 5.44E-04 | 32.941 | 9.5E-09 |
| rs6928363 | 6 | 152546094 | G | A | 0.17 (rs58415480) | 0.485 | 0.92 | 0.003 | 4.24E-04 | 46.849 | 7.7E-12 |
| rs58415480 | 6 | 152562271 | C | G | NA | 0.155 | 1.18 | −0.007 | 5.89E-04 | 124.672 | 6.0E-29 |
| rs75510204 | 6 | 152592680 | T | G | 0.07 (rs58415480) | 0.012 | 1.29 | −0.012 | 2.07E-03 | 32.664 | 1.1E-08 |
| rs6904757 | 6 | 152593102 | A | G | 0.08 (rs6928363) | 0.363 | 0.93 | 0.003 | 4.42E-04 | 41.035 | 1.5E-10 |
| rs144444583 | 6 | 152684585 | T | C | 0.30 (rs58415480) | 0.128 | 1.12 | −0.004 | 6.37E-04 | 45.286 | 1.7E-11 |
| rs138821078 | 9 | 674217 | C | G | 0.13 (rs10975820) | 0.021 | 1.23 | −0.008 | 1.50E-03 | 31.819 | 1.7E-08 |
| rs10975820 | 9 | 684160 | G | A | 0.00 (rs7027685) | 0.142 | 1.12 | −0.004 | 6.08E-04 | 48.602 | 3.1E-12 |
| rs7027685 | 9 | 802228 | A | T | NA | 0.402 | 1.11 | −0.004 | 4.33E-04 | 75.424 | 3.8E-18 |
| rs114680331 | 9 | 815682 | T | C | 0.12 (rs7027685) | 0.100 | 1.11 | −0.004 | 7.21E-04 | 35.577 | 2.5E-09 |
| rs4742448 | 9 | 826585 | C | G | 0.28 (rs7027685) | 0.458 | 1.07 | −0.003 | 4.33E-04 | 33.714 | 6.4E-09 |

*Appendix 1—table 2 continued*

| rs-code | Chr | Position | A | B | r² (reference SNP) | B freq | OR | Beta | SE | χ² | P |
|---------|-----|----------|---|---|---------------------|--------|-----|------|-----|-----|---|
| rs2277163 | 9 | 827224 | A | G | 0.07 (rs7027685) | 0.947 | 0.87 | 0.005 | 9.51E-04 | 33.211 | 8.3E-09 |
| rs1265164 | 10 | 105674854 | A | G | NA | 0.869 | 0.91 | 0.004 | 6.27E-04 | 32.772 | 1.0E-08 |
| rs11246003 | 11 | 213723 | T | G | 0.00 (rs507139) | 0.044 | 0.85 | 0.006 | 1.03E-03 | 32.337 | 1.3E-08 |
| rs507139 | 11 | 225196 | G | A | NA | 0.074 | 0.84 | 0.006 | 8.11E-04 | 53.082 | 3.2E-13 |
| rs2207548 | 11 | 32368744 | C | A | 0.24 (rs10835889) | 0.423 | 1.09 | −0.003 | 4.30E-04 | 62.360 | 2.9E-15 |
| rs10835889 | 11 | 32370380 | G | A | NA | 0.159 | 1.14 | −0.005 | 5.81E-04 | 79.234 | 5.5E-19 |
| rs7120483 | 11 | 32406983 | G | C | 0.30 (rs10835889) | 0.120 | 1.12 | −0.004 | 6.51E-04 | 44.481 | 2.6E-11 |
| rs11031783 | 11 | 32459923 | C | A | 0.28 (rs10835889) | 0.204 | 1.09 | −0.003 | 5.25E-04 | 36.687 | 1.4E-09 |
| rs2553772 | 11 | 35085453 | T | G | NA | 0.538 | 1.07 | −0.002 | 4.24E-04 | 34.458 | 4.4E-09 |
| rs59021565 | 11 | 107999907 | C | G | 0.27 (rs141379009) | 0.091 | 1.11 | −0.004 | 7.38E-04 | 30.882 | 2.7E-08 |
| rs141379009 | 11 | 108149207 | T | G | NA | 0.027 | 1.32 | −0.011 | 1.30E-03 | 76.719 | 2.0E-18 |
| rs4988023 | 11 | 108168995 | A | C | 0.00 (rs141379009) | 0.144 | 0.89 | 0.004 | 6.01E-04 | 43.699 | 3.8E-11 |
| rs12223381 | 11 | 108354102 | C | T | 0.12 (rs59021565) | 0.406 | 1.07 | −0.002 | 4.31E-04 | 30.234 | 3.8E-08 |
| rs9669403 | 12 | 46798900 | G | A | 0.28 (rs12832777) | 0.402 | 1.07 | −0.003 | 4.35E-04 | 36.998 | 1.2E-09 |
| rs12832777 | 12 | 46831129 | T | C | NA | 0.701 | 1.09 | −0.003 | 4.61E-04 | 44.734 | 2.3E-11 |
| rs117245733 | 13 | 40723944 | G | A | 0.00 (rs7986407) | 0.016 | 1.26 | −0.010 | 1.74E-03 | 34.519 | 4.2E-09 |
| rs7986407 | 13 | 41179798 | A | G | NA | 0.310 | 1.09 | −0.003 | 4.56E-04 | 45.943 | 1.2E-11 |
| rs66998222 | 16 | 51481596 | G | A | NA | 0.201 | 0.91 | 0.003 | 5.28E-04 | 42.053 | 8.9E-11 |
| rs78378222 | 17 | 7571752 | T | G | NA | 0.013 | 1.53 | −0.020 | 1.93E-03 | 105.457 | 9.7E-25 |
| rs733381 | 22 | 40669648 | A | G | NA | 0.213 | 1.10 | −0.003 | 5.16E-04 | 42.919 | 5.7E-11 |
| rs5936989 | X | 70022420 | T | A | 0.27 (rs5937008) | 0.782 | 0.92 | 0.005 | 9.30E-04 | 32.606 | 1.1E-08 |
| rs5937008 | X | 70093038 | C | T | NA | 0.520 | 0.91 | 0.006 | 7.68E-04 | 65.570 | 5.6E-16 |
| rs7059898 | X | 70149078 | C | A | 0.27 (rs5937008) | 0.359 | 1.07 | −0.005 | 8.11E-04 | 31.756 | 1.7E-08 |
| rs7888560 | X | 131171122 | A | G | 0.19 (rs5930554) | 0.228 | 1.08 | −0.005 | 9.16E-04 | 31.830 | 1.7E-08 |
| rs5930554 | X | 131312089 | T | C | NA | 0.311 | 1.14 | −0.009 | 8.29E-04 | 107.076 | 4.3E-25 |
| rs5933158 | X | 131578034 | A | G | 0.21 (rs5930554) | 0.586 | 1.08 | −0.005 | 7.98E-04 | 38.833 | 4.6E-10 |
| rs5975338 | X | 131626317 | A | G | 0.28 (rs5930554) | 0.129 | 1.10 | −0.006 | 1.14E-03 | 32.201 | 1.4E-08 |

DOI: https://doi.org/10.7554/eLife.37110.018

**Appendix 1—table 3.** Meta-analysis of the UKBB and Helsinki cohorts.

All the genome-wide significant, LD-independent ($r^2 \leq 0.3$) associations from the meta-analysis stage. Seven SNPs were LD-independent when compared to the discovery stage SNPs, and rs117245733 is shown for reference. The numbers for regression coefficients (Beta), standard error of beta (SE) and association (P) were collected from the UKBB and Helsinki summary statistics and their fixed effect model meta-analysis. The bolded SNP was the only one to reach a suggestive association ($p<10^{-5}$) in both cohorts.

| rs-code | Chr | Position | A | B | r² (reference SNP) | UKBB cohort | | | Helsinki cohort | | | Meta-analysis (fixed eff.) | | |
|---|---|---|---|---|---|---|---|---|---|---|---|---|---|---|
| | | | | | | Beta | SE | P | Beta | SE | P | Beta | SE | P |
| rs17631680 | 2 | 67090367 | T | C | NA | 0.004 | 0.0007 | 2.1E-07 | 0.005 | 0.0029 | 6.2E-02 | 0.004 | 0.0007 | 4.29E-08 |
| rs1735537 | 3 | 128122820 | T | C | NA | −0.003 | 0.0005 | 1.2E-07 | −0.004 | 0.0021 | 8.7E-02 | −0.003 | 0.0005 | 3.01E-08 |
| rs67751869 | 4 | 54568834 | T | C | 0.14 (rs62323680) | −0.005 | 0.0009 | 9.9E-08 | −0.011 | 0.0039 | 3.7E-03 | −0.005 | 0.0009 | 5.05E-09 |
| rs6901631 | 6 | 152567047 | T | C | 0.26 (rs6904757) | 0.003 | 0.0006 | 5.4E-08 | 0.005 | 0.0031 | 1.2E-01 | 0.003 | 0.0006 | 1.76E-08 |
| rs11790408 | 9 | 876418 | G | T | 0.10 (rs4742448) | 0.002 | 0.0004 | 6.5E-08 | 0.002 | 0.0019 | 3.9E-01 | 0.002 | 0.0004 | 4.89E-08 |
| rs117245733 | 13 | 40723944 | G | A | 0.00 (rs79986407) | −0.010 | 0.0017 | 4.2E-09 | −0.024 | 0.0053 | 8.1E-06 | −0.012 | 0.0017 | 3.18E-12 |
| rs10415391 | 19 | 22652436 | C | T | NA | −0.004 | 0.0007 | 2.8E-07 | −0.006 | 0.0028 | 2.0E-02 | −0.004 | 0.0007 | 2.87E-08 |
| rs62132801 | 19 | 49267882 | A | T | NA | 0.004 | 0.0007 | 3.3E-07 | 0.006 | 0.0029 | 2.8E-02 | 0.004 | 0.0007 | 4.20E-08 |

DOI: https://doi.org/10.7554/eLife.37110.019

**Appendix 1—table 4.** Genomic risk score. Summary of the GRS and its weights based on the discovery and meta-analysis stages. Dosage-based odds-ratios (OR) and log-odds were collected from the UKBB summary statistics. The A and B alleles are on GRCh37 forward strand, and the B allele is the effect allele. In total 50 SNPs from stage 1 and seven SNPs from stage 2.

| rs-code | Chr | Position | A | B | OR | Log-odds | Stage [*] |
|---------|-----|----------|---|---|-----|----------|--------|
| rs2235529 | 1 | 22450487 | C | T | 1.14 | 0.132 | Stage 1 |
| rs2092315 | 1 | 22507684 | C | T | 1.07 | 0.072 | Stage 1 |
| rs10929757 | 2 | 11702661 | A | C | 1.08 | 0.073 | Stage 1 |
| rs11674184 | 2 | 11721535 | T | G | 0.94 | −0.067 | Stage 1 |
| rs10936600 | 3 | 169514585 | A | T | 0.91 | −0.095 | Stage 1 |
| rs143835293 | 3 | 197623337 | A | G | 1.75 | 0.558 | Stage 1 |
| rs62323680 | 4 | 54546192 | G | A | 1.16 | 0.153 | Stage 1 |
| rs2202282 | 4 | 70634441 | C | T | 1.07 | 0.071 | Stage 1 |
| rs72709458 | 5 | 1283755 | C | T | 1.12 | 0.111 | Stage 1 |
| rs2736100 | 5 | 1286516 | C | A | 0.91 | −0.090 | Stage 1 |
| rs2853676 | 5 | 1288547 | T | C | 0.91 | −0.089 | Stage 1 |
| rs2456181 | 5 | 176450837 | C | G | 1.07 | 0.066 | Stage 1 |
| rs4870084 | 6 | 152543949 | C | T | 0.92 | −0.087 | Stage 1 |
| rs6928363 | 6 | 152546094 | G | A | 0.92 | −0.078 | Stage 1 |
| rs58415480 | 6 | 152562271 | C | G | 1.18 | 0.167 | Stage 1 |
| rs75510204 | 6 | 152592680 | T | G | 1.29 | 0.257 | Stage 1 |
| rs6904757 | 6 | 152593102 | A | G | 0.93 | −0.078 | Stage 1 |
| rs144444583 | 6 | 152684585 | T | C | 1.12 | 0.111 | Stage 1 |
| rs138821078 | 9 | 674217 | C | G | 1.23 | 0.205 | Stage 1 |
| rs10975820 | 9 | 684160 | G | A | 1.12 | 0.112 | Stage 1 |
| rs7027685 | 9 | 802228 | A | T | 1.11 | 0.103 | Stage 1 |
| rs114680331 | 9 | 815682 | T | C | 1.11 | 0.108 | Stage 1 |
| rs4742448 | 9 | 826585 | C | G | 1.07 | 0.066 | Stage 1 |
| rs2277163 | 9 | 827224 | A | G | 0.87 | −0.140 | Stage 1 |
| rs1265164 | 10 | 105674854 | A | G | 0.91 | −0.097 | Stage 1 |
| rs11246003 | 11 | 213723 | T | G | 0.85 | −0.167 | Stage 1 |
| rs507139 | 11 | 225196 | G | A | 0.84 | −0.171 | Stage 1 |
| rs2207548 | 11 | 32368744 | C | A | 1.09 | 0.091 | Stage 1 |
| rs10835889 | 11 | 32370380 | G | A | 1.14 | 0.133 | Stage 1 |
| rs7120483 | 11 | 32406983 | G | C | 1.12 | 0.112 | Stage 1 |
| rs11031783 | 11 | 32459923 | C | A | 1.09 | 0.084 | Stage 1 |
| rs2553772 | 11 | 35085453 | T | G | 1.07 | 0.069 | Stage 1 |
| rs59021565 | 11 | 107999907 | C | G | 1.11 | 0.109 | Stage 1 |
| rs141379009 | 11 | 108149207 | T | G | 1.32 | 0.281 | Stage 1 |
| rs4988023 | 11 | 108168995 | A | C | 0.89 | −0.113 | Stage 1 |
| rs12223381 | 11 | 108354102 | C | T | 1.07 | 0.064 | Stage 1 |
| rs9669403 | 12 | 46798900 | G | A | 1.07 | 0.071 | Stage 1 |
| rs12832777 | 12 | 46831129 | T | C | 1.09 | 0.086 | Stage 1 |
| rs117245733 | 13 | 40723944 | G | A | 1.26 | 0.231 | Stage 1 |
| rs7986407 | 13 | 41179798 | A | G | 1.09 | 0.084 | Stage 1 |

*Appendix 1—table 4 continued on next page*

Appendix 1—table 4 continued

| rs-code | Chr | Position | A | B | OR | Log-odds | Stage [*] |
|---|---|---|---|---|---|---|---|
| rs66998222 | 16 | 51481596 | G | A | 0.91 | −0.098 | Stage 1 |
| rs78378222 | 17 | 7571752 | T | G | 1.53 | 0.427 | Stage 1 |
| rs733381 | 22 | 40669648 | A | G | 1.10 | 0.091 | Stage 1 |
| rs5936989 | X | 70022420 | T | A | 0.92 | −0.081 | Stage 1 |
| rs5937008 | X | 70093038 | C | T | 0.91 | −0.094 | Stage 1 |
| rs7059898 | X | 70149078 | C | A | 1.07 | 0.068 | Stage 1 |
| rs7888560 | X | 131171122 | A | G | 1.08 | 0.077 | Stage 1 |
| rs5930554 | X | 131312089 | T | C | 1.14 | 0.129 | Stage 1 |
| rs5933158 | X | 131578034 | A | G | 1.08 | 0.074 | Stage 1 |
| rs5975338 | X | 131626317 | A | G | 1.10 | 0.094 | Stage 1 |
| rs17631680 | 2 | 67090367 | T | C | 0.90 | −0.102 | Stage 2 |
| rs1735537 | 3 | 128122820 | T | C | 1.07 | 0.071 | Stage 2 |
| rs67751869 | 4 | 54568834 | T | C | 1.13 | 0.121 | Stage 2 |
| rs6901631 | 6 | 152567047 | T | C | 0.91 | −0.097 | Stage 2 |
| rs11790408 | 9 | 876418 | G | T | 0.94 | −0.063 | Stage 2 |
| rs10415391 | 19 | 22652436 | C | T | 1.10 | 0.098 | Stage 2 |
| rs62132801 | 19 | 49267882 | A | T | 0.90 | −0.105 | Stage 2 |

[*] Discovered in stage 1 (UKBB GWAS) or in stage 2 (meta-analysis of UKBB and Helsinki)
DOI: https://doi.org/10.7554/eLife.37110.020

**Appendix 1—table 5.** Summary of association tests for SNPs. Each of the 57 GRS SNPs was tested for an additive effect to age at hysterectomy, degree of somatic allelic imbalance (AI) and tumor counts. Somatic allele imbalance was defined as the mean of the length of somatic allelic imbalance over all tumors of a patient. Tests of *MED12* mutation positive and negative tumors are denoted by MED12mut + and MED12mut-, respectively. The numbers for regression coefficient (Beta), standard error of beta (SE), test statistic (z) and association (P) were taken by fitting either a linear regression or negative binomial (NB) regression model (response ~predictor). The nominal P-values were adjusted for FDR (Q). The risk allele was used as the effect allele for Beta. In total 228 tests, all p<0.05 are shown.

| Predictor | Response | Model | Beta | SE | Statistic | P | Q |
|---|---|---|---|---|---|---|---|
| 12:46798900:G:A | MED12mut + count | NB | −0.22 | 0.08 | −2.88 | 0.004 | 0.51 |
| 9:674217:C:G | MED12mut + count | NB | −0.69 | 0.26 | −2.63 | 0.008 | 0.51 |
| X:70093038:C:T | MED12mut + count | NB | 0.21 | 0.08 | 2.62 | 0.009 | 0.51 |
| X:70022420:T:A | MED12mut + count | NB | 0.20 | 0.08 | 2.56 | 0.011 | 0.51 |
| 11:32459923:C:A | MED12mut + count | NB | 0.23 | 0.09 | 2.54 | 0.011 | 0.51 |
| 4:54568834:T:C | log(somatic AI basepairs) | Linear | −0.25 | 0.10 | −2.48 | 0.013 | 0.51 |
| 17:7571752:T:G | MED12mut- count | NB | −0.92 | 0.41 | −2.24 | 0.025 | 0.56 |
| 6:152546094:G:A | MED12mut- count | NB | −0.19 | 0.08 | −2.24 | 0.025 | 0.56 |
| 11:107999907:C:G | Age at hysterectomy | Linear | 3.00 | 1.34 | 2.24 | 0.026 | 0.56 |
| 13:40723944:G:A | MED12mut + count | NB | 0.36 | 0.17 | 2.17 | 0.030 | 0.56 |
| 22:40669648:A:G | MED12mut + count | NB | 0.19 | 0.09 | 2.15 | 0.031 | 0.56 |
| 5:1286516:C:A | MED12mut + count | NB | 0.17 | 0.08 | 2.15 | 0.031 | 0.56 |
| 22:40669648:A:G | Age at hysterectomy | Linear | 1.36 | 0.63 | 2.16 | 0.032 | 0.56 |
| 1:22450487:C:T | Age at hysterectomy | Linear | −1.48 | 0.72 | −2.05 | 0.041 | 0.65 |
| 16:51481596:G:A | log(somatic AI basepairs) | Linear | 0.17 | 0.08 | 2.02 | 0.044 | 0.65 |
| 5:1288547:T:C | Age at hysterectomy | Linear | 1.32 | 0.66 | 2.00 | 0.046 | 0.65 |

DOI: https://doi.org/10.7554/eLife.37110.021

**Appendix 1—table 6.** Summary of all the association tests for GRS. All GRS related tests from the main text. The notation of MED12mut + and MED12mut- refer to the numbers of *MED12*-mutation-positive and -negative tumors, respectively. The tests include Wilcoxon rank-sum and models for linear and negative binomial (NB) regression (variable ~GRS). The P values were adjusted for FWER with the Holm-Bonferroni method (Q). Significant associations (Q < 0.05) are shown bolded. Note that population association tests include only the female controls.

| GRS [*] | Cohort | Variable | Test | N cases | N controls | Rate ratio | P | Q |
|---|---|---|---|---|---|---|---|---|
| Stage 1 | Helsinki | UL phenotype | Rank-sum (one-tailed) | 457 | 8899 | - | 8.3e-10 | **1.1e-08** |
| Stage 2 | NFBC | UL phenotype | Rank-sum (one-tailed) | 459 | 2351 | - | 1.1e-05 | **1.1e-04** |
| Stage 2 | Helsinki | Total number of ULs | NB | 457 | - | 1.25 | 0.00105 | **0.0032** |
| Stage 2 | Helsinki | Age at hyster-ectomy | Linear | 392 | - | 0.50 | 0.48 | 0.48 |
| Stage 2 | Helsinki | Number of MED12mut+ | NB | 457 | - | 1.43 | 3.2e-04 | **0.002** |
| Stage 2 | Helsinki | Number of MED12mut- | NB | 457 | - | 0.79 | 0.0266 | 0.053 |
| Stage 2 | Helsinki | One-or-more MED12mut+ | Rank-sum | 334 | 123 | - | 5.3e-04 | **0.0026** |
| Stage 2 | Helsinki | All MED12mut+ | Rank-sum | 221 | 123 | - | 7.9e-04 | **0.0032** |
| Stage 2 | African [#] | UL phenotype | Rank-sum (one-tailed) | 296 | 1256 | - | 1.3e-05 | **1.2e-04** |
| Stage 2 | Caribbean [#] | UL phenotype | Rank-sum (one-tailed) | 668 | 2041 | - | 6.7e-05 | **5.4e-04** |
| Stage 2 | Irish [#] | UL phenotype | Rank-sum (one-tailed) | 398 | 6208 | - | 8.3e-06 | **9.1e-05** |
| Stage 2 | Indian [#] | UL phenotype | Rank-sum (one-tailed) | 203 | 2567 | - | 2.9e-04 | **0.0020** |
| Stage 2 | Other white [#] | UL phenotype | Rank-sum (one-tailed) | 647 | 8982 | - | 6.9e-09 | **8.3e-08** |

[*]Based either on stage 1 (UKBB GWAS) or stage 2 (meta-analysis of UKBB and Helsinki)
[#] An independent subset of UKBB data (stratified based on self-reported UKBB annotation).

DOI: https://doi.org/10.7554/eLife.37110.022

**Appendix 1—table 8.** eQTLs in tumor and normal tissue. Here, B is the effect allele. All local permutation p<0.05 are shown. Full table of eQTLs is given in *Supplementary file 4*.

| Tissue | Gene | ID | N | Best SNP | Distance | Nominal P | Permutation P | FDR |
|--------|------|-----|---|----------|----------|-----------|---------------|-----|
| T | RP11-816J6.3 | ENSG00000269889.1 | 82 | rs67795055 | 42457 | 9.10E-05 | 4.03E-04 | 0.224 |
| T | RUFY3 | ENSG00000018189.8 | 8 | rs7660770 | −972945 | 7.84E-04 | 6.68E-04 | 0.370 |
| T | TRIP13 | ENSG00000071539.9 | 22 | rs2736099 | 394581 | 1.63E-04 | 2.27E-03 | 1 |
| N | RN7SL832P | ENSG00000243819.3 | 28 | rs7570979 | 886958 | 4.06E-04 | 2.75E-03 | 1 |
| N | LNX1 | ENSG00000072201.9 | 25 | rs62323674 | 212300 | 7.12E-04 | 3.22E-03 | 1 |
| N | RP11-849F2.8 | ENSG00000269928.1 | 5 | rs143094271 | −665038 | 2.43E-03 | 4.00E-03 | 1 |
| T | MIR5006 | ENSG00000264190.1 | 158 | rs9549254 | −907660 | 6.10E-04 | 4.42E-03 | 1 |
| N | RP11-423H2.3 | ENSG00000249684.1 | 50 | rs183686 | −883235 | 3.36E-03 | 5.55E-03 | 1 |
| N | SLC7A3 | ENSG00000165349.7 | 148 | rs4360450 | 965 | 6.75E-04 | 7.38E-03 | 1 |
| T | ALPL | ENSG00000162551.9 | 52 | . | 521370 | 8.48E-04 | 7.59E-03 | 1 |
| N | PRRG4 | ENSG00000135378.3 | 167 | rs11031737 | −478718 | 3.99E-04 | 8.59E-03 | 1 |
| N | RP11-791G15.2 | ENSG00000272275.1 | 28 | rs6432220 | 802714 | 1.24E-03 | 8.81E-03 | 1 |
| T | ZDHHC11B | ENSG00000206077.6 | 22 | rs2736099 | 576864 | 9.74E-04 | 1.02E-02 | 1 |
| T | ACAP1 | ENSG00000072818.7 | 5 | rs78378222 | 331903 | 4.83E-03 | 1.04E-02 | 1 |
| N | HNRNPA1P48 | ENSG00000224578.3 | 20 | rs55881068 | −212266 | 4.97E-03 | 1.12E-02 | 1 |
| T | WNT4 | ENSG00000162552.10 | 52 | rs12042083 | 28933 | 1.30E-03 | 1.16E-02 | 1 |
| T | LRRIQ4 | ENSG00000188306.6 | 82 | rs13074500 | 25860 | 2.96E-03 | 1.20E-02 | 1 |
| T | TEX11 | ENSG00000120498.9 | 148 | rs11795627 | 208650 | 1.30E-03 | 1.23E-02 | 1 |
| T | AC011994.1 | ENSG00000265583.1 | 28 | rs11685032 | −73236 | 1.84E-03 | 1.32E-02 | 1 |
| T | FRMD7 | ENSG00000165694.5 | 215 | rs5933158 | 367012 | 8.40E-04 | 1.33E-02 | 1 |
| N | TRAPPC1 | ENSG00000170043.7 | 5 | rs138420351 | −133601 | 7.23E-03 | 1.44E-02 | 1 |
| T | RP11-423H2.1 | ENSG00000170089.11 | 72 | rs353496 | −806511 | 9.37E-03 | 1.51E-02 | 1 |
| N | SDHAP3 | ENSG00000185986.10 | 22 | rs11742908 | −297655 | 1.42E-03 | 1.58E-02 | 1 |
| T | CDC42-IT1 | ENSG00000230068.2 | 52 | rs10917167 | 118400 | 2.10E-03 | 1.80E-02 | 1 |
| N | RP11-362K14.6 | ENSG00000270096.1 | 82 | rs12638862 | −35245 | 4.49E-03 | 1.96E-02 | 1 |
| T | TNRC6B | ENSG00000100354.16 | 32 | rs12484951 | 262253 | 8.29E-03 | 1.97E-02 | 1 |
| N | DNAH2 | ENSG00000183914.10 | 5 | rs143094271 | −157571 | 9.73E-03 | 1.98E-02 | 1 |
| N | SALL1 | ENSG00000103449.7 | 20 | rs12933686 | 308913 | 9.53E-03 | 2.13E-02 | 1 |
| T | XXyac-YRM2039.2 | ENSG00000181404.13 | 71 | rs138821078 | 659705 | 1.69E-03 | 2.22E-02 | 1 |
| N | ITPRIP | ENSG00000148841.11 | 8 | rs9419958 | −395949 | 1.43E-02 | 2.39E-02 | 1 |
| T | RNASEK-C17orf49 | ENSG00000161939.14 | 5 | . | 501685 | 1.19E-02 | 2.43E-02 | 1 |
| T | UQCRFS1P1 | ENSG00000226085.2 | 32 | rs1807579 | 256926 | 1.06E-02 | 2.44E-02 | 1 |
| T | FOXO1 | ENSG00000150907.6 | 166 | rs9549260 | 124299 | 2.94E-03 | 2.61E-02 | 1 |
| T | NDUFS6 | ENSG00000145494.7 | 22 | rs7734992 | −521387 | 2.81E-03 | 2.82E-02 | 1 |
| T | RP11-259O2.3 | ENSG00000249731.1 | 22 | rs7710703 | −680704 | 2.89E-03 | 2.84E-02 | 1 |
| T | CDC42 | ENSG00000070831.11 | 52 | rs41307810 | −25076 | 3.35E-03 | 2.91E-02 | 1 |

*Appendix 1—table 8 continued on next page*

**Appendix 1—table 7.** Population specific estimates for GRS.
Allele frequencies were collected from gnomAD (version 2.0.1). GRS weights were taken from UKBB summary statistics. The estimate for population specific GRS follows from weighting the allele frequencies with log-odds. The final values were scaled with a factor two to make them comparable with SNP dosage-based GRS. See *Appendix-figure 6* for an explanation of the population acronyms.

| rs-code | Chr | Pos | A | B | AF_AFR | AF_AMR | AF_ASJ | AF_EAS | AF_FIN | AF_NFE | AF_OTH |
|---|---|---|---|---|---|---|---|---|---|---|---|
| rs58415480 | 6 | 152562271 | C | G | 0.228 | 0.066 | 0.090 | 0.000 | 0.237 | 0.161 | 0.158 |
| rs78378222 | 17 | 7571752 | T | G | 0.002 | 0.004 | 0.000 | 0.000 | 0.020 | 0.018 | 0.015 |
| rs10835889 | 11 | 32370380 | G | A | 0.384 | 0.091 | 0.202 | 0.052 | 0.069 | 0.150 | 0.124 |
| rs141379009 | 11 | 108149207 | T | G | 0.005 | 0.014 | 0.023 | 0.000 | 0.007 | 0.022 | 0.015 |
| rs7027685 | 9 | 802228 | A | T | 0.535 | 0.602 | 0.357 | 0.738 | 0.556 | 0.413 | 0.462 |
| rs2235529 | 1 | 22450487 | C | T | 0.029 | 0.313 | 0.046 | 0.484 | 0.164 | 0.158 | 0.171 |
| rs72709458 | 5 | 1283755 | C | T | 0.139 | 0.137 | 0.183 | 0.134 | 0.221 | 0.188 | 0.180 |
| rs2207548 | 11 | 32368744 | C | A | 0.553 | 0.228 | 0.393 | 0.179 | 0.321 | 0.411 | 0.378 |
| rs2736100 | 5 | 1286516 | C | A | 0.550 | 0.597 | 0.477 | 0.606 | 0.507 | 0.504 | 0.539 |
| rs507139 | 11 | 225196 | G | A | 0.028 | 0.058 | 0.079 | 0.007 | 0.094 | 0.072 | 0.084 |
| rs62323680 | 4 | 54546192 | G | A | 0.020 | 0.032 | 0.099 | 0.002 | 0.045 | 0.070 | 0.060 |
| rs2853676 | 5 | 1288547 | T | C | 0.765 | 0.801 | 0.613 | 0.856 | 0.783 | 0.739 | 0.763 |
| rs10975820 | 9 | 684160 | G | A | 0.669 | 0.397 | 0.212 | 0.586 | 0.257 | 0.171 | 0.255 |
| rs10936600 | 3 | 169514585 | A | T | 0.094 | 0.406 | 0.195 | 0.539 | 0.274 | 0.249 | 0.267 |
| rs6928363 | 6 | 152546094 | G | A | 0.692 | 0.693 | 0.541 | 0.851 | 0.575 | 0.508 | 0.568 |
| rs7986407 | 13 | 41179798 | A | G | 0.446 | 0.459 | 0.277 | 0.739 | 0.384 | 0.311 | 0.367 |
| rs144444583 | 6 | 152684585 | T | C | 0.050 | 0.045 | 0.044 | 0.006 | 0.230 | 0.139 | 0.155 |
| rs143835293 | 3 | 197623337 | A | G | 0.000 | 0.004 | 0.003 | 0.000 | 0.015 | 0.002 | 0.006 |
| rs12832777 | 12 | 46831129 | T | C | 0.918 | 0.740 | 0.762 | 0.902 | 0.651 | 0.675 | 0.669 |
| rs7120483 | 11 | 32406983 | G | C | 0.242 | 0.087 | 0.152 | 0.025 | 0.074 | 0.112 | 0.098 |
| rs4988023 | 11 | 108168995 | A | C | 0.022 | 0.044 | 0.142 | 0.006 | 0.236 | 0.146 | 0.149 |
| rs733381 | 22 | 40669648 | A | G | 0.095 | 0.471 | 0.169 | 0.247 | 0.258 | 0.216 | 0.239 |

*Appendix 1—table 7 continued*

| rs-code | Chr | Pos | A | B | AF_AFR | AF_AMR | AF_ASJ | AF_EAS | AF_FIN | AF_NFE | AF_OTH |
|---|---|---|---|---|---|---|---|---|---|---|---|
| rs66998222 | 16 | 51481596 | G | A | 0.033 | 0.138 | 0.265 | 0.001 | 0.166 | 0.213 | 0.167 |
| rs6904757 | 6 | 152593102 | A | G | 0.443 | 0.579 | 0.401 | 0.625 | 0.379 | 0.392 | 0.401 |
| rs2202282 | 4 | 70634441 | C | T | 0.230 | 0.549 | 0.420 | 0.726 | 0.437 | 0.491 | 0.466 |
| rs10929757 | 2 | 11702661 | A | C | 0.116 | 0.505 | 0.556 | 0.517 | 0.674 | 0.591 | 0.556 |
| rs9669403 | 12 | 46798900 | G | A | 0.615 | 0.476 | 0.406 | 0.266 | 0.396 | 0.382 | 0.369 |
| rs11031783 | 11 | 32459923 | C | A | 0.262 | 0.364 | 0.288 | 0.748 | 0.197 | 0.204 | 0.241 |
| rs114680331 | 9 | 815682 | T | C | 0.065 | 0.178 | 0.067 | 0.128 | 0.190 | 0.115 | 0.133 |
| rs117245733 | 13 | 40723944 | G | A | 0.001 | 0.004 | 0.003 | 0.000 | 0.030 | 0.024 | 0.027 |
| rs2553772 | 11 | 35085453 | T | G | 0.342 | 0.656 | 0.600 | 0.769 | 0.524 | 0.544 | 0.569 |
| rs2456181 | 5 | 176450837 | C | G | 0.796 | 0.355 | 0.533 | 0.481 | 0.500 | 0.511 | 0.505 |
| rs4742448 | 9 | 826585 | C | G | 0.838 | 0.629 | 0.473 | 0.785 | 0.548 | 0.471 | 0.503 |
| rs2277163 | 9 | 827224 | A | G | 0.959 | 0.858 | 0.964 | 0.825 | 0.966 | 0.948 | 0.954 |
| rs4870084 | 6 | 152543949 | C | T | 0.100 | 0.421 | 0.175 | 0.188 | 0.234 | 0.211 | 0.238 |
| rs1265164 | 10 | 105674854 | A | G | 0.469 | 0.891 | 0.825 | 0.985 | 0.890 | 0.863 | 0.873 |
| rs75510204 | 6 | 152592680 | T | G | 0.002 | 0.002 | 0.000 | 0.000 | 0.015 | 0.011 | 0.011 |
| rs11246003 | 11 | 213723 | T | G | 0.060 | 0.047 | 0.033 | 0.149 | 0.022 | 0.035 | 0.038 |
| rs138821078 | 9 | 674217 | C | G | 0.002 | 0.004 | 0.000 | 0.000 | 0.017 | 0.025 | 0.014 |
| rs59021565 | 11 | 107999907 | C | G | 0.038 | 0.112 | 0.119 | 0.049 | 0.050 | 0.086 | 0.054 |
| rs2092315 | 1 | 22507684 | C | T | 0.132 | 0.323 | 0.242 | 0.504 | 0.200 | 0.243 | 0.210 |
| rs12223381 | 11 | 108354102 | C | T | 0.413 | 0.606 | 0.477 | 0.444 | 0.390 | 0.414 | 0.457 |
| rs11674184 | 2 | 11721535 | T | G | 0.259 | 0.476 | 0.334 | 0.414 | 0.403 | 0.384 | 0.389 |
| rs5930554 | X | 131312089 | T | C | 0.752 | 0.174 | 0.296 | 0.104 | 0.320 | 0.319 | 0.311 |
| rs5937008 | X | 70093038 | C | T | 0.145 | 0.383 | 0.414 | 0.563 | 0.486 | 0.520 | 0.485 |
| rs5933158 | X | 131578034 | A | G | 0.710 | 0.588 | 0.500 | 0.686 | 0.650 | 0.605 | 0.741 |

*Appendix 1—table 7 continued on next page*

Appendix 1—table 7 continued

| rs-code | Chr | Pos | A | B | AF_AFR | AF_AMR | AF_ASJ | AF_EAS | AF_FIN | AF_NFE | AF_OTH |
|---|---|---|---|---|---|---|---|---|---|---|---|
| rs5936989 | X | 70022420 | T | A | 0.742 | 0.888 | 0.767 | 0.994 | 0.677 | 0.778 | 0.810 |
| rs5975338 | X | 131626317 | A | G | 0.409 | 0.103 | 0.099 | 0.091 | 0.161 | 0.145 | 0.151 |
| rs7059898 | X | 70149078 | C | A | 0.143 | 0.167 | 0.355 | 0.001 | 0.340 | 0.342 | 0.276 |
| rs7888560 | X | 131171122 | A | G | 0.142 | 0.090 | 0.199 | 0.000 | 0.244 | 0.217 | 0.201 |
| rs67751869 | 4 | 54568834 | T | C | 0.085 | 0.030 | 0.099 | 0.022 | 0.053 | 0.062 | 0.067 |
| rs6901631 | 6 | 152567047 | T | C | 0.256 | 0.249 | 0.182 | 0.122 | 0.102 | 0.120 | 0.127 |
| rs10415391 | 19 | 22652436 | C | T | 0.202 | 0.074 | 0.110 | 0.225 | 0.130 | 0.107 | 0.144 |
| rs1735537 | 3 | 128122820 | T | C | 0.523 | 0.385 | 0.262 | 0.425 | 0.249 | 0.251 | 0.288 |
| rs62132801 | 19 | 49267882 | A | T | 0.013 | 0.070 | 0.053 | 0.009 | 0.112 | 0.103 | 0.092 |
| rs17631680 | 2 | 67090367 | T | C | 0.018 | 0.061 | 0.050 | 0.001 | 0.100 | 0.102 | 0.082 |
| rs11790408 | 9 | 876418 | G | T | 0.110 | 0.307 | 0.387 | 0.246 | 0.355 | 0.380 | 0.353 |
| | | | | | AFR | AMR | ASJ | EAS | FIN | NFE | OTH |
| GRS | | | | | 4.718 | 4.059 | 4.068 | 4.196 | 4.168 | 4.091 | 4.116 |

DOI: https://doi.org/10.7554/eLife.37110.023

*Appendix 1—table 8 continued*

| Tissue | Gene | ID | N | Best SNP | Distance | Nominal P | Permutation P | FDR |
|---|---|---|---|---|---|---|---|---|
| T | RP3-363L9.1 | ENSG00000227064.1 | 215 | rs3764771 | −830073 | 2.03E-03 | 2.97E-02 | 1 |
| T | RP11-362K14.7 | ENSG00000270135.1 | 82 | rs67795055 | 18611 | 7.45E-03 | 3.07E-02 | 1 |
| N | AMIGO2 | ENSG00000139211.5 | 155 | rs9706162 | −606808 | 6.28E-03 | 3.08E-02 | 1 |
| T | LPIN1 | ENSG00000134324.7 | 28 | rs11685032 | −143427 | 5.06E-03 | 3.33E-02 | 1 |
| T | MBNL3 | ENSG00000076770.10 | 215 | rs1263155 | 190718 | 2.55E-03 | 3.37E-02 | 1 |
| N | TOX3 | ENSG00000103460.12 | 5 | rs12325192 | −983789 | 3.24E-02 | 3.38E-02 | 1 |
| N | DANCR | ENSG00000226950.2 | 23 | rs62325482 | 885823 | 8.24E-03 | 3.54E-02 | 1 |
| T | DMRT1 | ENSG00000137090.7 | 71 | rs12004436 | −158261 | 3.09E-03 | 3.64E-02 | 1 |
| N | NT5C2 | ENSG00000076685.14 | 8 | rs1265164 | 828913 | 2.05E-02 | 3.75E-02 | 1 |
| T | TMEM256-PLSCR3 | ENSG00000187838.12 | 5 | rs138420351 | 407016 | 1.77E-02 | 3.81E-02 | 1 |
| T | RP11-401P9.4 | ENSG00000261685.2 | 20 | rs12933686 | 799079 | 1.64E-02 | 3.84E-02 | 1 |
| N | CTC1 | ENSG00000178971.9 | 5 | rs143094271 | −667090 | 1.89E-02 | 4.03E-02 | 1 |
| N | SULT1B1 | ENSG00000173597.4 | 19 | rs13133166 | 13318 | 1.49E-02 | 4.04E-02 | 1 |
| N | TNK1 | ENSG00000174292.8 | 5 | rs143094271 | 179248 | 2.24E-02 | 4.21E-02 | 1 |
| T | SOX15 | ENSG00000129194.3 | 5 | rs138420351 | 208566 | 2.02E-02 | 4.30E-02 | 1 |
| T | UGT2A1 | ENSG00000173610.7 | 19 | rs1587766 | 112959 | 2.06E-02 | 4.31E-02 | 1 |
| T | RP4-607I7.1 | ENSG00000255521.1 | 34 | rs2553783 | −82085 | 2.66E-02 | 4.41E-02 | 1 |
| N | SENP3 | ENSG00000161956.8 | 5 | rs78378222 | 106559 | 2.19E-02 | 4.45E-02 | 1 |
| N | KDM6B | ENSG00000132510.6 | 5 | rs143094271 | −280121 | 2.16E-02 | 4.54E-02 | 1 |
| T | SKIL | ENSG00000136603.9 | 82 | rs34194057 | −525916 | 1.13E-02 | 4.63E-02 | 1 |
| T | EPHB2 | ENSG00000133216.12 | 52 | . | −680104 | 6.04E-03 | 4.97E-02 | 1 |

DOI: https://doi.org/10.7554/eLife.37110.024

**Appendix 1—table 9.** cis-meQTLs in candidate genes where FDR < 0.1. The statistics were calculated with an additive linear model. Here B is used as the effect allele. Full Table of cis-meQTLs with nominal significance (p-value<0.05) is given in *Supplementary file 5*.

| Tissue | SNP | CpG* | Beta | t-stat | p-value | FDR | Gene | CpG island | Promoter | AA | AB | BB |
|---|---|---|---|---|---|---|---|---|---|---|---|---|
| N | 13_41179798 | 13_41189143 | −0.413 | −19.837 | 6.57E-20 | 4.88E-15 | FOXO1 | no | no | 15 | 16 | 4 |
| T | 13_41179798 | 13_41189143 | −0.333 | −8.296 | 3.29E-11 | 7.74E-07 | FOXO1 | no | no | 28 | 23 | 5 |
| T | 5_1283755 | 5_1285974 | −0.427 | −7.167 | 2.21E-09 | 3.46E-05 | TERT | no | no | 36 | 17 | 3 |
| N | 5_1283755 | 5_1285974 | −0.461 | −7.46 | 1.42E-08 | 1.11E-04 | TERT | no | no | 18 | 14 | 3 |
| T | 5_1286516 | 5_1277576 | −0.289 | −6.358 | 4.53E-08 | 3.99E-04 | TERT | yes | no | 7 | 29 | 20 |
| T | 1_22450487 | 1_22456326 | −0.123 | −5.507 | 1.04E-06 | 6.36E-03 | WNT4 | no | no | 40 | 15 | 1 |
| N | 5_1286516 | 5_1285974 | 0.426 | 5.477 | 4.51E-06 | 1.76E-02 | TERT | no | no | 6 | 22 | 7 |
| T | 5_1286516 | 5_1285974 | 0.299 | 4.739 | 1.61E-05 | 4.82E-02 | TERT | no | no | 7 | 29 | 20 |
| N | 13_41179798 | 13_41223110 | −0.174 | −4.957 | 2.10E-05 | 5.36E-02 | FOXO1 | no | no | 15 | 16 | 4 |
| N | 5_1283755 | 5_1282413 | −0.17 | −4.816 | 3.16E-05 | 7.12E-02 | TERT | no | no | 18 | 14 | 3 |
| T | 11_225196 | 11_195431 | 0.222 | 4.452 | 4.31E-05 | 9.21E-02 | BET1L | no | ODF3 | 48 | 8 | 0 |

* Minimum CpG coverage of ≥2 required in ≥90% of samples. CpG in 1 Mb flank from the SNP.
DOI: https://doi.org/10.7554/eLife.37110.025

**Appendix 1—table 10.** Replication of previously suggested UL predisposition loci. For study details, see the literature references in the main text. The population, locus, gene and risk allele (RA) information were collected from the original studies. The associations, P (UKBB) column, were taken from the UKBB summary statistics for 15,453 UL cases. The bolded values pass FWER (Bonferroni for seven independent loci; p<0.05/7).

| Study | Population | Locus | Suggested gene | SNP | RA | Method | P (UKBB) |
|---|---|---|---|---|---|---|---|
| Cha et al. | Japanese | 22q13.1 | TNRC6B | rs12484776 | G | GWAS | **1.0E-10** |
| Edwards et al. | European Americans | 22q13.1 | TNRC6B | rs12484776 | G | GWAS | **1.0E-10** |
| Cha et al. | Japanese | 11p15.5 | BET1L | rs2280543 | G | GWAS | **2.2E-08** |
| Edwards et al. | European Americans | 11p15.5 | BET1L | rs2280543 | G | GWAS | **2.2E-08** |
| Zhang et al. | African Americans | 1q42.2 | PCNXL2 | rs7546784 | - | Admixture | 3.6E-02 |
| Zhang et al. | African Americans | 2q32.2 | PMS1 | rs256552 | - | Admixture | 1.8E-01 |
| Cha et al. | Japanese | 10q24.33 | SLK | rs7913069 | A | GWAS | 3.4E-01 |
| Hellwege et al. | African American | 22q13.1 | CYTH4 | rs5995416 | C | GWAS | 5.3E-01 |
| Hellwege et al. | African American | 22q13.1 | CYTH4 | rs739187 | C | GWAS | 6.2E-01 |
| Hellwege et al. | African American | 22q13.1 | CYTH4 | rs713939 | C | GWAS | 8.0E-01 |
| Hellwege et al. | African American | 22q13.1 | CYTH4 | rs4821628 | G | GWAS | 8.1E-01 |
| Eggert et al. | Multiple | 17q25.3 | FASN | rs4247357 | A | Linkage and GWAS | 8.1E-01 |

DOI: https://doi.org/10.7554/eLife.37110.026

**Appendix 1—table 11.** GWAS catalog and references to earlier literature on GRS SNPs. The GRS SNPs rs10936600, rs11674184, rs2235529, rs2736100, rs2853676, rs72709458 and rs78378222 were found in the GWAS catalog (version 1.0.1; www.ebi.ac.uk/gwas). The numbers for allele frequency (AF), odds-ratio (OR) and association (P) follow those reported in the GWAS catalog.

| Pubmed | Date | Journal | | Gene symbol | Risk allele | AF | OR | P |
|---|---|---|---|---|---|---|---|---|
| 18835860 | 2008-10-01 | J Med Genet | Idiopathic pulmonary fibrosis | TERT | rs2736100-A | 0.41 | 2.11 | 3.00E-08 |
| 19578367 | 2009-07-05 | Nat Genet | Glioma | TERT | rs2736100-G | 0.49 | 1.27 | 2.00E-17 |
| 19578367 | 2009-07-05 | Nat Genet | Glioma | TERT | rs2853676-A | 0.73 | 1.26 | 4.00E-14 |
| 19836008 | 2009-10-15 | Am J Hum Genet | Lung adenocarcinoma | TERT | rs2736100-G | 0.5 | 1.12 | 2.00E-10 |
| 20139978 | 2010-02-07 | Nat Genet | Red blood cell count | TERT | rs2736100-G | 0.4 | 0.07 | 3.00E-08 |
| 20543847 | 2010-06-13 | Nat Genet | Testicular germ cell cancer | TERT | rs2736100-T | 0.49 | 1.33 | 8.00E-15 |
| 20700438 | 2010-08-05 | PLoS Genet | Lung adenocarcinoma | TERT | rs2736100-G | 0.39 | 1.46 | 2.00E-22 |

*Appendix 1—table 11 continued on next page*

*Appendix 1—table 11 continued*

| Pubmed | Date | Journal | | Gene symbol | Risk allele | AF | OR | P |
|---|---|---|---|---|---|---|---|---|
| 20871597 | 2010-09-26 | Nat Genet | Lung adenocarcinoma | TERT | rs2736100-C | 0.39 | 1.27 | 3.00E-11 |
| 21531791 | 2011-04-29 | Hum Mol Genet | Glioma | TERT | rs2736100-? | - | 1.25 | 1.00E-14 |
| 21725308 | 2011-07-03 | Nat Genet | Lung cancer | TERT | rs2736100-C | 0.41 | 1.27 | 1.00E-27 |
| 21827660 | 2011-08-09 | BMC Med Genomics | Glioma | TERT | rs2736100-? | - | - | 7.00E-09 |
| 21946351 | 2011-09-25 | Nat Genet | Basal cell carcinoma | TP53 | rs78378222-C | - | 2.16 | 2.00E-20 |
| 22886559 | 2012-08-11 | Hum Genet | Glioma | TERT | rs2736100-G | 0.494 | 1.30 | 4.00E-09 |
| 23143601 | 2012-11-11 | Nat Genet | Lung cancer | TERT | rs2736100-G | 0.4 | 1.38 | 4.00E-27 |
| 23472165 | 2013-03-05 | PLoS One | Endometriosis | WNT4 | rs2235529-T | 0.152 | 1.28 | 7.00E-09 |
| 23472165 | 2013-03-05 | PLoS One | Endometriosis | WNT4 | rs2235529-C | 0.709 | 1.19 | 6.00E-06 |
| 23472165 | 2013-03-05 | PLoS One | Endometriosis | WNT4 | rs2235529-T | 0.134 | 1.25 | 8.00E-07 |
| 23472165 | 2013-03-05 | PLoS One | Endometriosis | WNT4 | rs2235529-A | 0.153 | 1.30 | 3.00E-09 |
| 23583980 | 2013-04-14 | Nat Genet | Interstitial lung disease | TERT | rs2736100-A | 0.49 | 1.37 | 2.00E-19 |
| 24403052 | 2014-01-08 | Hum Mol Genet | Basal cell carcinoma | TP53 | rs78378222-G | - | 2.24 | 4.00E-22 |
| 24465473 | 2014-01-21 | PLoS One | Telomere length | TERT | rs2736100-C | | 0.08 | 4.00E-06 |
| 24908248 | 2014-06-08 | Nat Genet | Glioma (high-grade) | TERT | rs2736100-C | 0.51 | 1.39 | 1.00E-15 |
| 24945404 | 2014-06-19 | PLoS Genet | Bone mineral density (paediatric, upper limb) | WNT4 | rs2235529-C | 0.85 | 0.12 | 1.00E-08 |
| 24945404 | 2014-06-19 | PLoS Genet | Bone mineral density (paediatric, upper limb) | WNT4 | rs2235529-C | - | 0.12 | 3.00E-07 |
| 25855136 | 2015-04-09 | Nat Commun | Basal cell carcinoma | TP53 | rs78378222-G | 0.018 | 2.07 | 1.00E-20 |
| 26424050 | 2015-10-01 | Nat Commun | Glioblastoma | - | rs72709458-T | - | 1.68 | 6.00E-24 |
| 27363682 | 2016-07-01 | Nat Commun | Multiple myeloma | - | rs10936600-A | - | 1.20 | 6.00E-15 |
| 27501781 | 2016-08-09 | Nat Commun | EGFR mutation-positive lung adenocarcinoma | TERT | rs2736100-G | 0.387 | 1.42 | 2.00E-31 |
| 27539887 | 2016-08-19 | Nat Commun | Basal cell carcinoma | TP53 | rs78378222-G | 0.01 | 1.41 | 2.00E-10 |
| 27863252 | 2016-11-17 | Cell | Mean corpuscular hemoglobin | TP53 | rs78378222-G | 0.0121 | 0.10 | 6.00E-09 |
| 27863252 | 2016-11-17 | Cell | Mean corpuscular hemoglobin | TERT | rs2736100-A | 0.4982 | 0.04 | 5.00E-34 |

*Appendix 1—table 11 continued on next page*

*Appendix 1—table 11 continued*

| Pubmed | Date | Journal | | Gene symbol | Risk allele | AF | OR | P |
|--------|------|---------|---|-------------|-------------|-----|-----|---|
| 27863252 | 2016-11-17 | Cell | Platelet count | TERT | rs2736100-A | 0.4984 | 0.03 | 3.00E-20 |
| 28017375 | 2016-12-22 | Am J Hum Genet | Mean corpuscular volume | TERT | rs2736100-A | - | 0.00 | 3.00E-06 |
| 28017375 | 2016-12-22 | Am J Hum Genet | Mean corpuscular volume | TERT | rs2736100-? | - | - | 2.00E-11 |
| 28017375 | 2016-12-22 | Am J Hum Genet | Mean corpuscular hemoglobin | TERT | rs2736100-? | - | - | 1.00E-08 |
| 28135244 | 2017-01-30 | Nat Genet | Pulse pressure | TP53 | rs78378222-T | 0.99 | 0.90 | 2.00E-10 |
| 28346443 | 2017-03-27 | Nat Genet | Glioma | TP53 | rs78378222-G | 0.013 | 2.53 | 9.00E-38 |
| 28346443 | 2017-03-27 | Nat Genet | Glioblastoma | TP53 | rs78378222-G | 0.013 | 2.63 | 5.00E-29 |
| 28346443 | 2017-03-27 | Nat Genet | Non-glioblastoma glioma | TP53 | rs78378222-G | 0.013 | 2.73 | 5.00E-27 |
| 28537267 | 2017-05-24 | Nat Commun | Endometriosis | GREB1 | rs11674184-T | 0.61 | 1.13 | 3.00E-17 |
| 28537267 | 2017-05-24 | Nat Commun | Endometriosis | GREB1 | rs11674184-T | 0.61 | 1.12 | 3.00E-14 |
| 28604728 | 2017-06-12 | Nat Genet | Testicular germ cell tumor | TERT | rs2736100-A | 0.51 | 1.28 | 9.00E-25 |
| 28604732 | 2017-06-12 | Nat Genet | Testicular germ cell tumor | TERT | rs2736100-A | 0.5 | 1.29 | 8.00E-20 |

DOI: https://doi.org/10.7554/eLife.37110.027

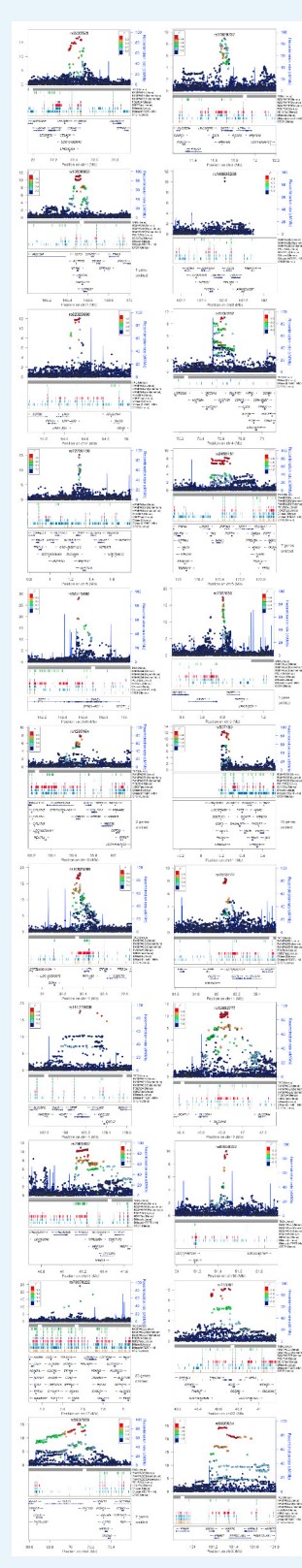

**Appendix 1—figure 1.** Structure of UL predisposition loci. The lead SNPs from stage one and their genomic context. The LD estimates ($r^2$) were taken from UK10k ALSPAC. Associations are based on the UKBB cohort. Gene symbols and ENCODE tracks (details in Supplementary

Methods) are shown for reference; coordinates follow hg19. See main text *Table 1* for more information. In total 22 figures, ordered by genomic position.
DOI: https://doi.org/10.7554/eLife.37110.028

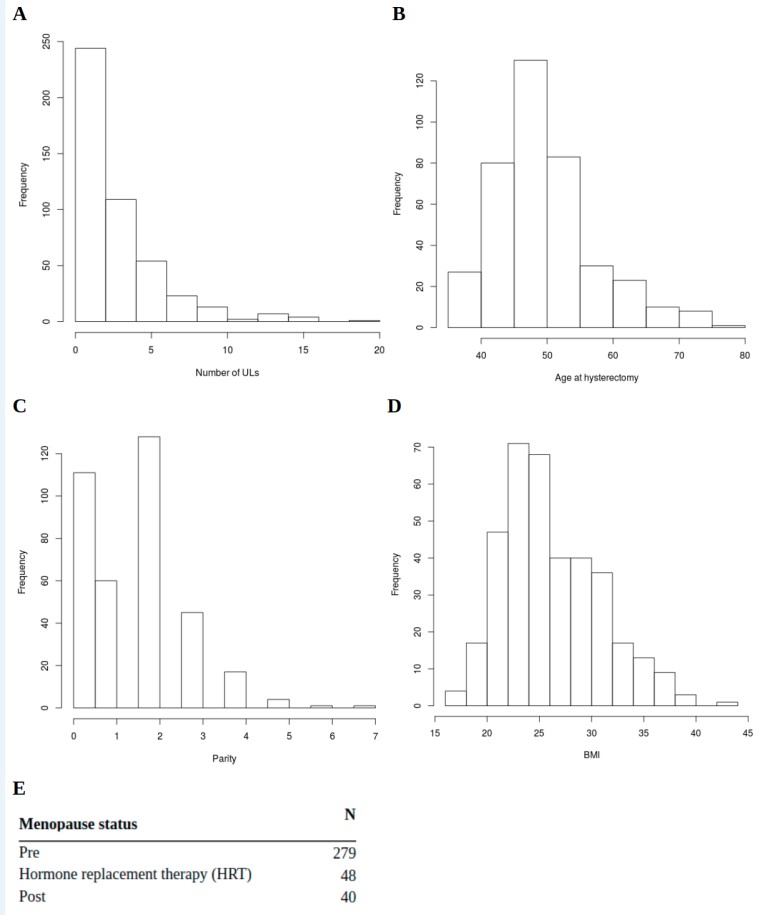

**Appendix 1—figure 2.** Clinical background information for the Helsinki cohort patients. (**A,**) the number of ULs per patient (n = 457). (**B**), patient's age at hysterectomy (n = 392). (**C**), parity (n = 367). (**D**), body mass index (BMI; n = 366). (**E**), menopause status (n = 367).
DOI: https://doi.org/10.7554/eLife.37110.029

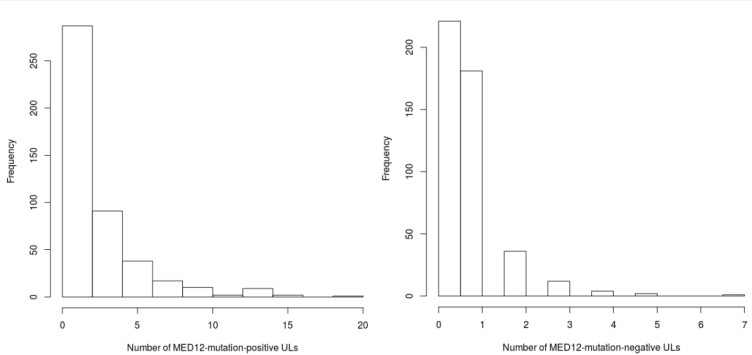

**Appendix 1—figure 3.** *MED12* mutation status distributions for the Helsinki cohort patients. On left, mutation-positive tumors per patient (n = 457), and on right, mutation-negative tumors per patient (n = 457).
DOI: https://doi.org/10.7554/eLife.37110.030

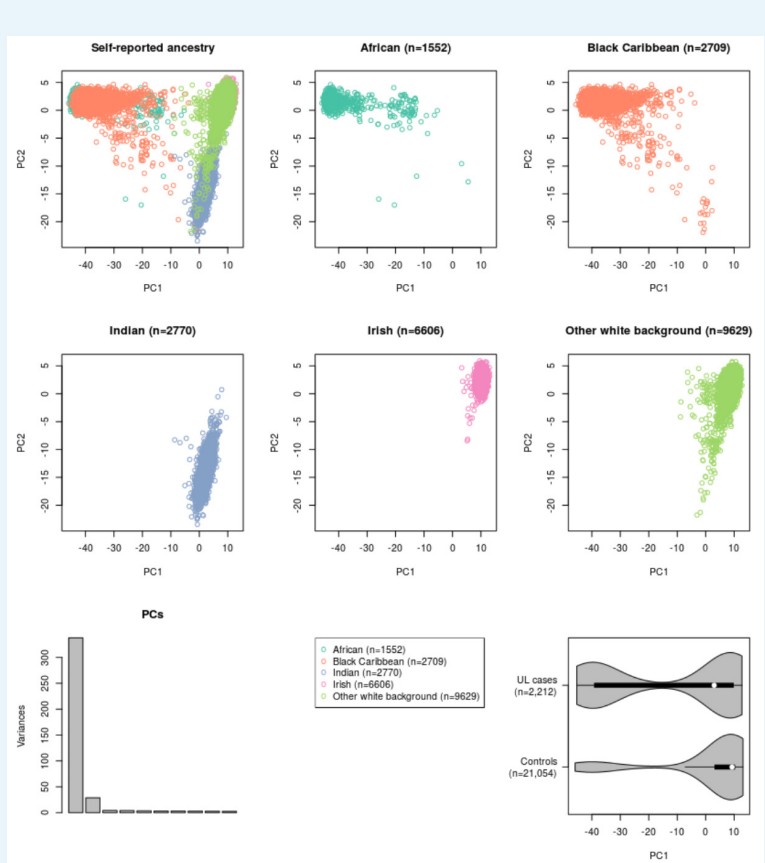

**Appendix 1—figure 4.** Genetic separation between the self-reported ancestries in UK Biobank. Principal components (PC) analysis of ancestry informative markers. In total 1396 autosomal, ancestry informative SNPs were used and the resulting first two PCs are shown: top-left plot shows all 23,266 samples colored according to their self-reported ancestry. The subsequent plots show the same data divided by the self-reported ancestry. Bottom-left plot displays the variance explained by the first ten PCs. Bottom-right violin plot displays the distribution of the first principal component for cases (n = 2,212) and controls (n = 21,054). The phenotype is more prevalent in individuals with increased African ancestry ($p < 2.2 \times 10^{-16}$; Wilcoxon rank sum $W = 3 \times 10^7$).

DOI: https://doi.org/10.7554/eLife.37110.031

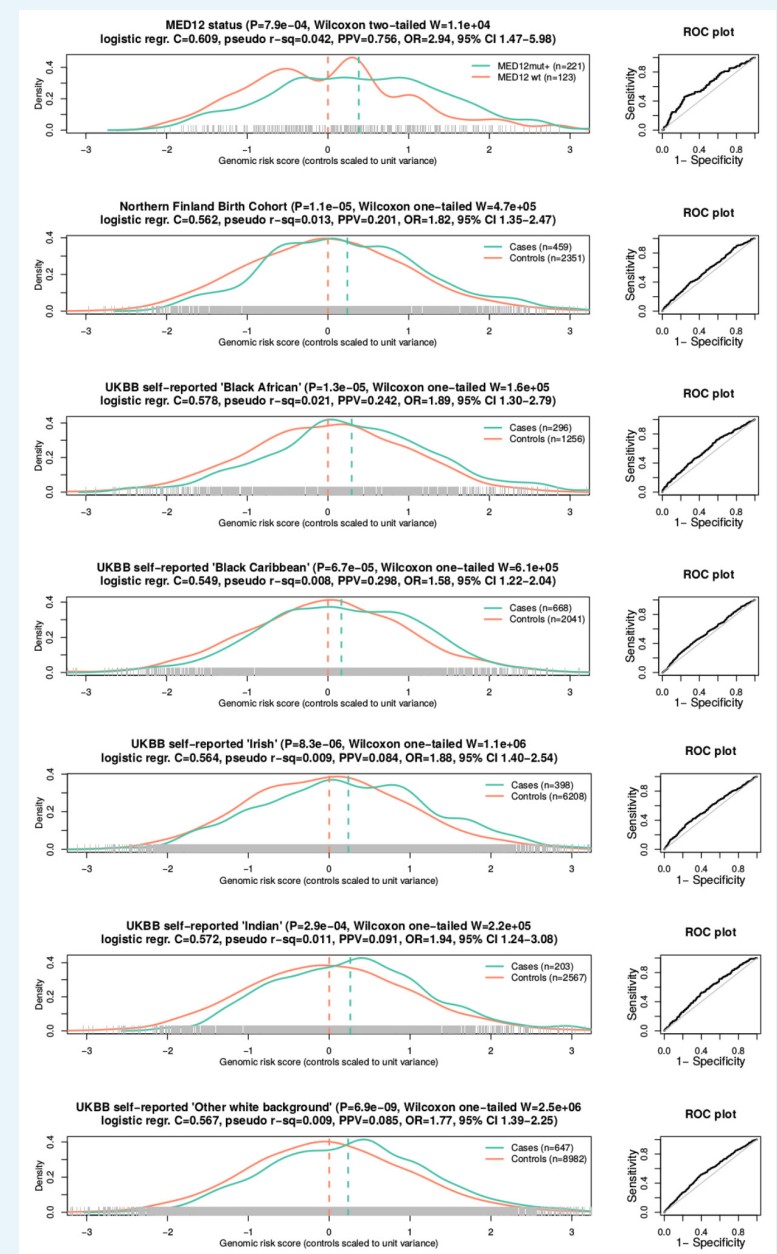

**Appendix 1—figure 5.** Association between GRS and phenotype. One somatic phenotype (*MED12* mutation status) and six independent case-control replication cohorts. On left, density plots (bandwidth = 0.2) for each phenotype. X-axis gives GRS values scaled to unit variance with respect to the controls. Dashed lines denote the mean GRS for cases and controls. Associations (P) and test statistics (W) are from Wilcoxon rank-sum tests. C-scores, Nagelkerke's $R^2$ (pseudo $R^2$) and Receiver operating characteristic (ROC; on right) are from a logistic regression model. Positive predictive values (PPV) were computed from the highest GRS quartile, and odds ratios (OR and 95% CI) from the top and bottom GRS quartiles.
DOI: https://doi.org/10.7554/eLife.37110.032

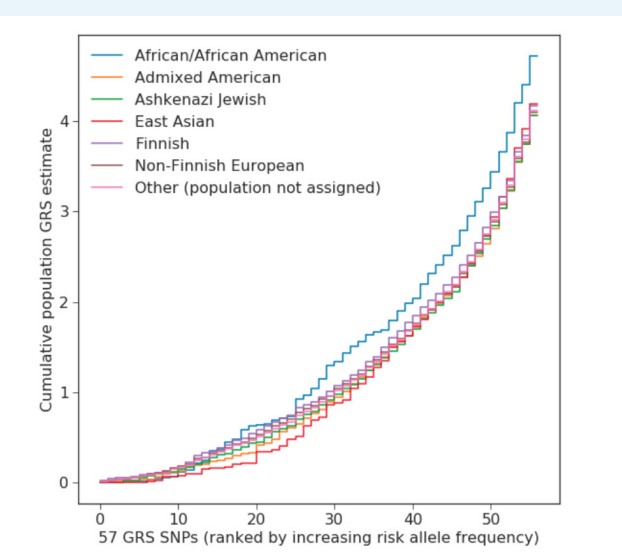

**Appendix 1—figure 6.** Genomic risk scores in gnomAD populations. Population-specific genomic risk scores (GRS) as seen based on gnomAD allele frequencies. Y-axis shows the cumulative effect of each of the 57 GRS SNPs; the X-axis is ordered by risk allele frequency. The population names displayed here follow the naming convention of the gnomAD database.

DOI: https://doi.org/10.7554/eLife.37110.033

**A**

```
glm.nb(formula = Total.number.of.myomas ~ GRS, data = a, init.theta =
2.979790747, link = log)

Deviance Residuals:
    Min       1Q   Median       3Q      Max
-1.3813  -1.0113  -0.4079   0.3739   3.0408

Coefficients:
            Estimate Std. Error z value Pr(>|z|)
(Intercept)  0.18987    0.30325   0.626  0.53124
GRS          0.22515    0.06874   3.275  0.00105 **
---
Signif. codes:  0 '***' 0.001 '**' 0.01 '*' 0.05 '.' 0.1 ' ' 1
(Dispersion parameter for Negative Binomial(2.9798) family taken to be 1)
    Null deviance: 436.03  on 456  degrees of freedom
Residual deviance: 425.34  on 455  degrees of freedom
AIC: 2011.5
Number of Fisher Scoring iterations: 1
            Theta:  2.980
        Std. Err.:  0.358
 2 x log-likelihood:  -2005.487
```

**B**

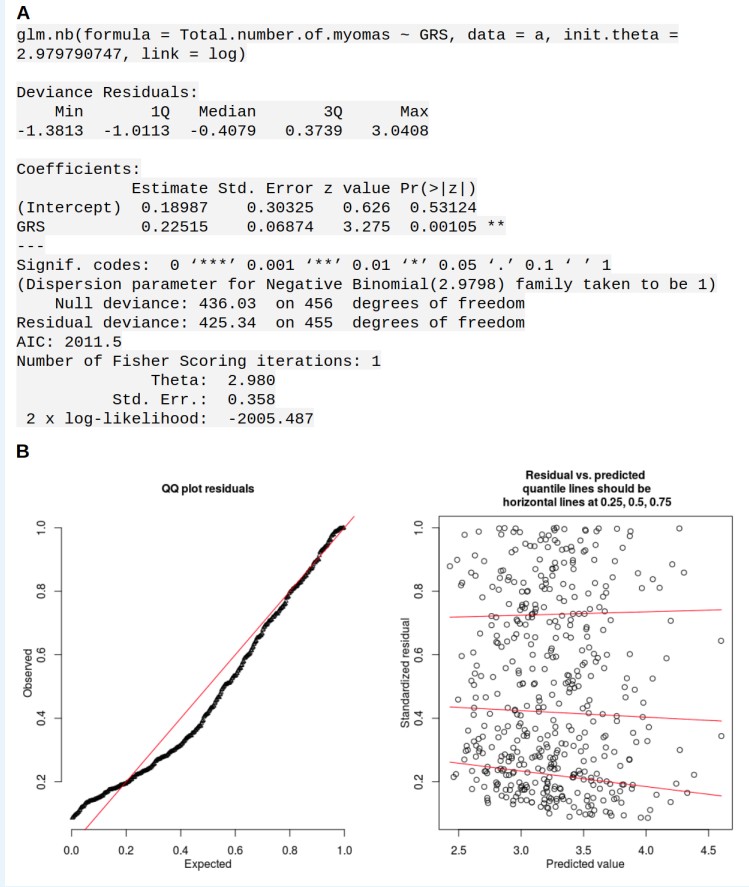

**Appendix 1—figure 7.** Association between GRS and number of ULs per patient. The

numbers are based on the Helsinki cohort data of 457 patients with all distinct tumors of ≥1 cm diameter harvested at hysterectomy; details on sample collection are given in Supplementary Methods. **A**, summary of the model. **B**, diagnostic plots for the model.
DOI: https://doi.org/10.7554/eLife.37110.034

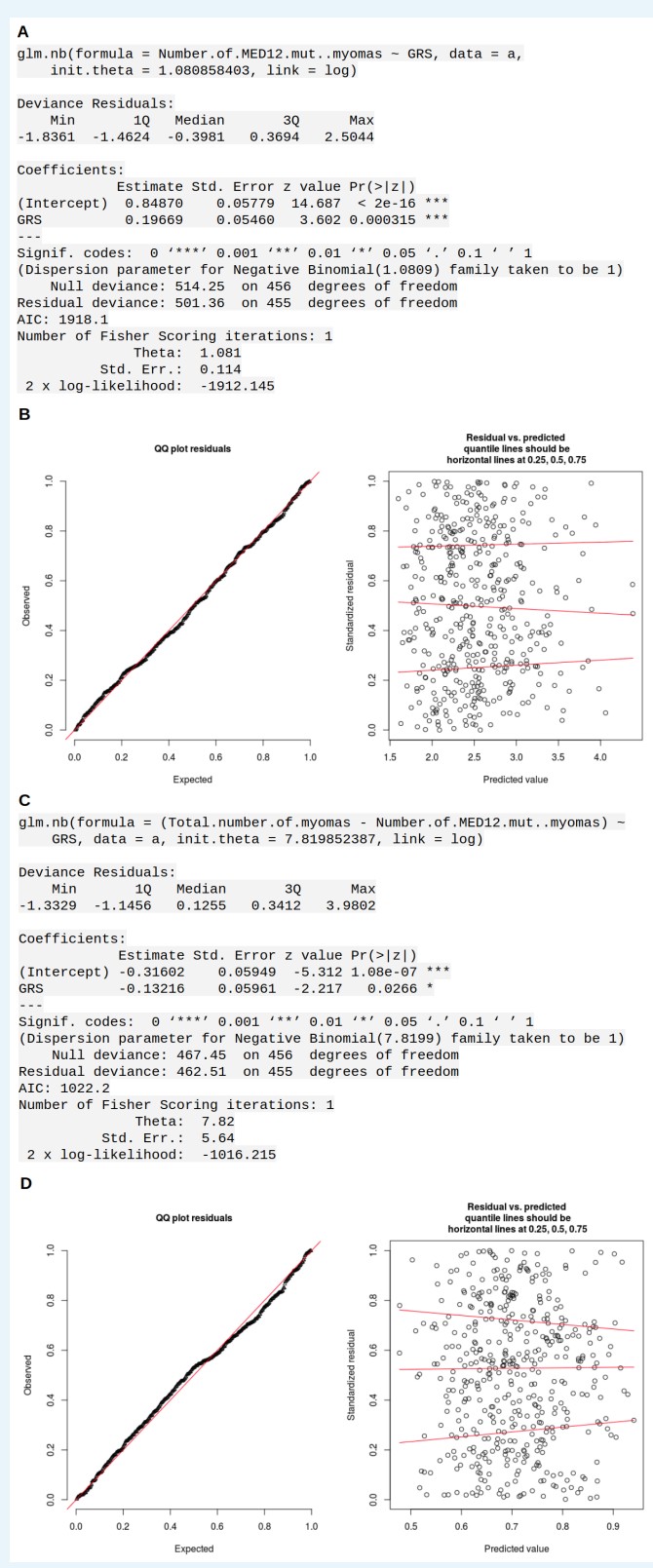

**Appendix 1—figure 8.** Association between GRS and *MED12* mutations. The numbers are based on the Helsinki cohort data (457 patients). (**A**) summary of the negative binomial model for *MED12*-mutation-positive tumor counts. (**B**) diagnostic plots for the negative binomial model. (**C, D**) similar model for the *MED12*-mutation-negative tumor counts.

DOI: https://doi.org/10.7554/eLife.37110.035

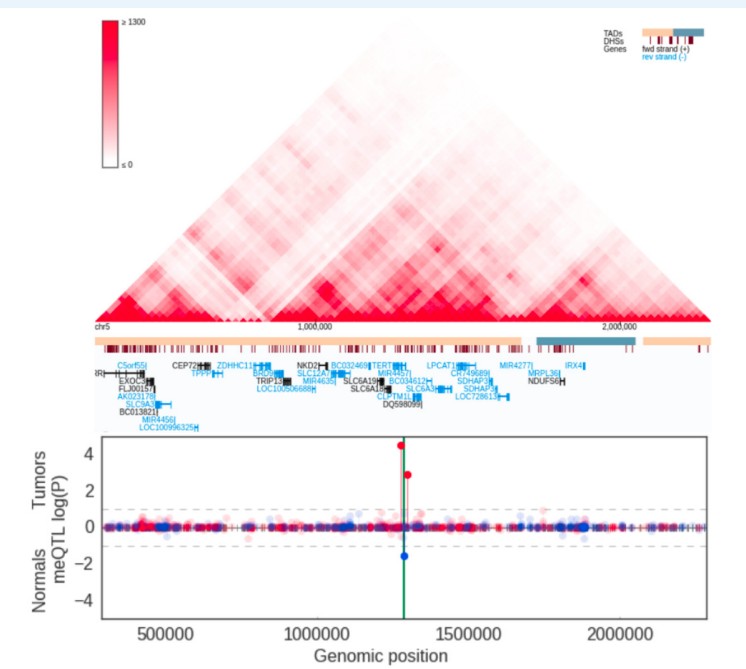

**Appendix 1—figure 9.** meQTLs at *TERT* region. Hi-C, TADs and CpG methylation around rs2736100 with an 1Mbp flank. The needle plot shows the meQTL associations (dashed lines at 10% FDR; green line denotes the two SNPs; gray ticks denote all CpGs tested; blue needle for positive coefficient, red for negative coefficient) for tumors (above x-axis; $n_{AA}$ = 8, $n_{AB}$ = 32, $n_{BB}$ = 16) and normals (below x-axis; $n_{AA}$ = 5, $n_{AB}$ = 22, $n_{BB}$ = 7). The only significant meQTLs were seen around *TERT*.

DOI: https://doi.org/10.7554/eLife.37110.036

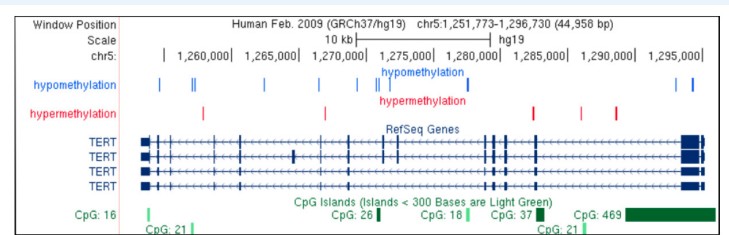

**Appendix 1—figure 10.** Overview of the CpG methylation around *TERT* in tumor tissue. There are several CpGs in *TERT* whose methylation level is associated with rs2736100 genotype (nominal p<0.05). Some of them are also detected in normal tissue (*Appendix 1— Table 9*). Hypomethylation refers to decreased methylation in BB vs. AA genotype and hypermethylation the opposite.

DOI: https://doi.org/10.7554/eLife.37110.037

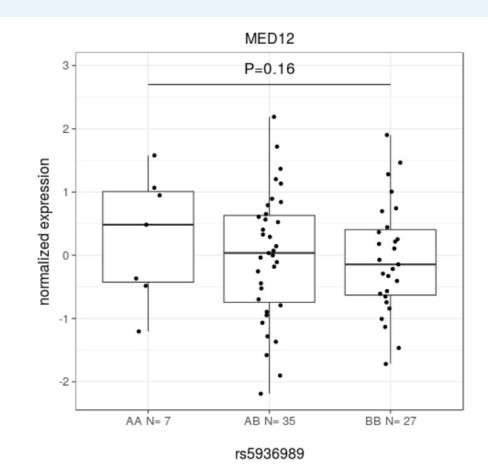

**Appendix 1—figure 11.** *MED12* expression. Linear model. Here, A is the risk allele. Beta 0.247 ± 0.177.

DOI: https://doi.org/10.7554/eLife.37110.038

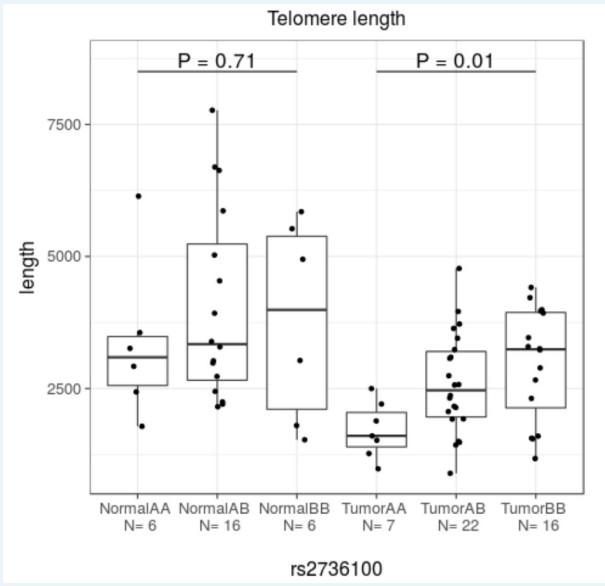

**Appendix 1—figure 12.** Telomere lengths in tumors and normals. Here A denotes the risk allele (see *Appendix 1—Table 1* for the allele information). In tumors, shorter telomere length is significantly associated with the risk allele at 5p15.33 (rs2736100) (Kruskal-Wallis test).

DOI: https://doi.org/10.7554/eLife.37110.039

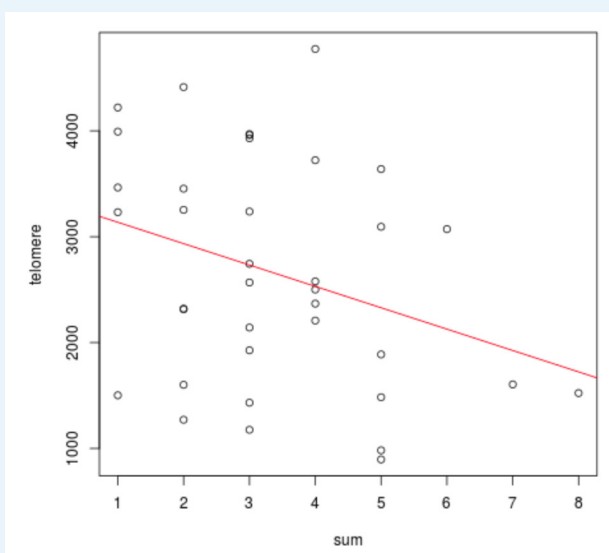

**Appendix 1—figure 13.** The association between risk allele count and telomere length On X-axis, total number of risk alleles at *TERT* (rs72709458, rs2736100, rs2853676), *TERC* (rs10936600) and *OBFC1* (rs1265164). On Y-axis, estimated telomere length. Linear regression model p=0.055; 95% CI −408.511 – 4.704 per one risk allele.

DOI: https://doi.org/10.7554/eLife.37110.040

