## [Decision Letter]

Thank you for submitting your article "Genetic predisposition to uterine leiomyoma is determined by loci for genitourinaty development and genome stability" for consideration by *eLife*. Your article has been reviewed by three peer reviewers, and the evaluation has been overseen by Eduardo Franco as the Reviewing Editor and Charles Sawyers as the Senior Editor.

As a key feature of *eLife*'s review process, the reviewers have discussed your paper and their critiques with one another in an online forum moderated by the Reviewing Editor and the Senior Editor. This process fosters consensus and provides additional guidance to assist authors in improving their work. Although your submission elicited much interest, our decision is that it needs major revisions before we can consider it for publication. We have captured in this decision letter the main points in the reviewers' critiques and in the subsequent discussion. Please keep in mind that the revised version may be shown to the reviewers of the original submission.

Summary:

Välimäki et al. performed a genome-wide association study (GWAS) of uterine leiomyomas with 15,453 cases and 392,628 controls to identify common genetic variants associated with uterine fibroids. In a discovery-phase GWAS across white women from the UK Biobank and Helsinki cohorts, the authors identify 22 independent loci to have significant associations with the phenotype of interest. Based on LD-pruning (r2 < 0.3), a set of 57 representative genetic variants were selected for quantifying the overall polygenic risk score/genomic risk score (GRS) in subsequent analyses. Rather than replicating the individual loci's associations with uterine fibroids, the authors demonstrate the reproducibility of elevated composite GRS scores in cases compared to controls in several analyses stratified by ancestry, clinical features, and *MED12* mutation status. Follow-up functional studies in the manuscript are reported for gene expression, DNA methylation, and telomere length association analyses. The authors conclude that their reported observations indicate a genetic predisposition to uterine fibroids, a heritable component to *MED12* somatic mutations in uterine fibroids, and an explanation for the increased prevalence of genitourinary disease in women of African origin.

Essential revisions:

1) Conclusions are not sufficiently justified by the findings:

This is a very interesting study that leverages several powerful resources to evaluate the relationship between GWAS variants and risk for uterine leiomyoma. This GWAS is poised to be the largest GWAS of uterine fibroids yet published. The authors identify several GWAS variants in discovery stage, but unfortunately, the effects did not replicate in stage 3 analyses. They did find strong associations between their polygenic risk score and uterine leiomyoma outcome and some promising findings from their secondary functional evaluates via tissue gene expression, DNA methylation, and telomere length analyses.

Primary concerns are with regard to the rigor and completeness of the post-GWAS studies. Current expectations of human genetics research require sufficient follow-up analysis to provide more comprehensive insight into the biological correlates of genetic associations. The numerous directions the authors took in the post-GWAS analyses made the paper lack a sense of synthesis or cohesion. There were a lot of analyses that were performed but the write up lacked focus and contained statements that were not by results. The reviewers recommended a more in-depth interpretation of findings and additional analyses (e.g., network or pathway analysis), in addition to tidying up the results.

Although the authors pursue a number of different follow-up analyses, none of the analyses are yet carried to completion, and therefore none are sufficiently mature to provide a strong, cohesive functional explanation of how the genes they have identified influence the risk for uterine fibroids. A more well-developed knowledge of the functional role of the associated loci in uterine fibroids will be an exciting addition to understanding the biology of these common neoplasms.

2) Group comparisons and other methodological details:

The authors do not provide sufficient detail on the tumor tissue experiments in the main text or how their experiments were performed. Some of these details are presented in the supplementary material but a minimal amount of additional details should be provided in the main text to allow readers to assess the overall quality of the analyses. Also, the supplementary materials describe the QC and processing details but details are missing regarding the statistical analyses of these tumor tissue studies.

Many of their functional analyses (e.g., gene expression, telomere length, and DNA methylation) involve comparing uterine fibroid samples to 'control' samples that are not well-defined. The authors should indicate whether the control tissue is patient-matched unaffected, adjacent myometrium or myometrial tissue isolated from a completely unaffected control.

In the discussion of gene expression analyses, the authors mention accessing 'myometrium' tissue on the GTEx Portal (Discussion, fourth paragraph). Currently, GTEx only provides data taken from the entire uterus. Moreover, given the nature of the individuals from whom GTEx tissue is derived (i.e., predominantly older individuals dying of natural causes), the relevance of these analyses to uterine gene expression in reproductive-aged women is questionable.

3) Statistical analysis:

In general the “Statistical analysis” Materials and methods section is not presented with sufficient detail in the main text.

Why did the author's perform Bonferroni corrections of the GWAS but FDR on other analyses? Would those analyses survive a Bonferroni Multiple testing correction?

GWAS meta-analyses across all three stages should be presented including heterogeneity statistics.

4) Ancestry and risk:

The authors use self-reported ancestry in their follow-up analyses. PCA-based analysis of ancestry-informative markers is necessary for ensuring homogeneity of the individual follow-up cohorts. The discovery and stage 2 meta-analyses were performed primarily in white European populations and did not replicate in the final stage 3 that included multiple ancestries, including those of African ancestry. The author's main conclusions are that their findings explain the occurrence of leiomyoma in women of African origin (both in the abstract and final discussion). Correlation between increased percentage of African ancestry and probability of reporting uterine fibroids is a necessary analysis for the conclusion that "[Their] findings… in part explain the more frequent occurrence of UL in women of African origin."

The authors should describe the frequencies of the variants evaluated in stage 2 and 3 for their respective populations. Is it possible that some of the variants that associated in stage 1 in European ancestry individuals are rare in other ancestral groups?

The interchanging usage of 'African American', 'Admixed American', and 'African' in the text should be addressed.

5) Correlations of findings with biology:

Recent work investigating the role of the *WNT4/CDC42* locus in endometriosis has shown that *CDC42* – not *WNT4* – contributes to the endometriosis phenotype. In light of the evidence drawn from analyses in endometriosis, the work presented in this manuscript provides insufficient evidence to implicate *WNT4* in uterine fibroids. Additional work to either verify or strengthen the specific implication of *WNT4* (from the *CDC42/WNT4* locus) is necessary, else the revised manuscript should more fully justify the conclusions around *WNT4*.

A more thorough and solid functional validation to prove the conclusions should be performed. Given the high frequency of *MED12* mutations in leiomyomas the heritability to *MED12* mutations is possibly one of the strongest points of the manuscript, however no real validation has been demonstrated in the study. The authors show a "trend" of an over-expression which cannot be taken as enough evidence for such a claim. What was the effect size in patients with both *MED12* mutated positive and negative leiomyoma? A strong association between the risk allele rs5937008 and the *MED12* status is described and the authors argue that the risk allele could facilitate selection of somatic *MED12* mutations, possibly through increased expression.

UL can be differentiated in 3 mutually exclusive subtypes, with the *MED12* mutant subtype been the most common. In the manuscript the authors refer and analyze carefully associations between risk alleles and *MED12* mutation positive samples, however there is no mention to the other two subtypes. From the set of 1481 lesions included in the study, what was the distribution of tumors and patients with FH deficient or *HMGA2* overexpressing lesions? Were FH mutation carriers included in the study? Was there a risk allele from the 22 significant loci that associated to any of these subtypes?

Although I understand that not all associations can be experimentally proven, a functional evaluation of this phenomenon and how would the rs5937008 drive *MED12* mutations would greatly improve the manuscript and provide strong evidence of the rs5937008 allele and UL development. These data would be of great importance towards clinical management of patients.

*TERT* SNPs rs72709458; rs2736100; rs2853676 are known to be related to many different neoplasias. How did those relate to *MED12* mutated tumors?

In the same fashion that for rs5937008, it would be of great interest to confirm a functional effect by which rs58415480 regulates *ESR1* expression or the repercussion of the lead SNP at 2p on *GREB1* mediated regulation of *ESR1*.

The authors divide the 22 risk loci into two global biological processes one of which is genomic instability and telomere maintenance. Genetic instability is a common mechanism driving tumorigenesis and molecular signatures of genomic instability can be extrapolated from tumor profiling, in this context, a molecular profile of the leiomyomas associated with *ATM, TP53, TERT, TERC* or *OBFC1* could reveal strong evidence to validate the role of genetic instability in UL development.

6) Additional issues in interpretation:

The follow-up analyses rely solely on the composite GRS as a surrogate measurement for genetic association with the phenotype of interest. It is assumed that this limited metric is selected due to the underpowered nature of the small follow-up cohorts.

None of the results presented in the 'Association to Gene Expression' section are actually significant. The authors should consider removing this section from the main portion of the manuscript. Similarly, the 'negative trend' observed in the 'combined effect of *TERT, TERC*, and *OBFC1*' on telomere length is also insignificant. The fact that the combined effect of three major genes still is insignificant indicates some concern with interpretation of the results.

The authors state that they replicated in NFBC and in the other ethnic groups but the replication results for the other five ethnic groups were not in the supplementary material. Those should be provided. In addition, the authors state (subsection “Replication of the GWAS and GRS”) that the third stage replicated the results but later on in the same paragraph state that no associations passed multiple testing corrections for the single SNP associations. The authors should state clearly that one of the associations replicated in stage 3, based on the evidence they present.

There are multiple locations where the authors state that results significantly associated but later on state that the results did not pass multiple testing corrections. The authors should make it clear that the there was no evidence for significant association if effects did not pass multiple testing corrections. This language is also present in the Discussion, where results are described as replicating in stage 3.

---

## [Author Response]

Essential revisions:1) Conclusions are not sufficiently justified by the findings:[…] Primary concerns are with regard to the rigor and completeness of the post-GWAS studies. Current expectations of human genetics research require sufficient follow-up analysis to provide more comprehensive insight into the biological correlates of genetic associations. The numerous directions the authors took in the post-GWAS analyses made the paper lack a sense of synthesis or cohesion. There were a lot of analyses that were performed but the write up lacked focus and contained statements that were not by results. The reviewers recommended a more in-depth interpretation of findings and additional analyses (e.g., network or pathway analysis), in addition to tidying up the results.Although the authors pursue a number of different follow-up analyses, none of the analyses are yet carried to completion, and therefore none are sufficiently mature to provide a strong, cohesive functional explanation of how the genes they have identified influence the risk for uterine fibroids. A more well-developed knowledge of the functional role of the associated loci in uterine fibroids will be an exciting addition to understanding the biology of these common neoplasms.

We thank the reviewers for their encouraging comments and have now improved the presentation throughout the manuscript as requested. We put specific emphasis on making the follow-up analyses easier to follow and unified the writing. Our study indeed extends well beyond a traditional GWAS. We combine population, patient and tumor derived resources in order to give a truly unique view into the hereditary risk of ULs. The wide-range of our post-GWAS analyses will serve as an exceptional basis for the scientific community to design future, in-depth functional studies to better characterize and develop models for UL tumorigenesis.

The reviewers suggested an additional pathway-based analysis to help the interpretation of the risk loci. We followed this recommendation by applying the DEPICT framework (Pers et al., 2015), which is designed to integrate multiple GWAS loci for *in silico* target gene prioritization, pathway enrichment and tissue-specific expression profiling. For analysis details, see the main text and supplementary material. In brief, applying DEPICT on the set of genome-wide significant SNPs did not reveal any significant enrichment with the exception of one pathway related to induced stress. *ATM* was the highest-ranking target gene, and uterus/myometrium were among the highest-ranking tissue types.

2) Group comparisons and other methodological details:The authors do not provide sufficient detail on the tumor tissue experiments in the main text or how their experiments were performed. Some of these details are presented in the supplementary material but a minimal amount of additional details should be provided in the main text to allow readers to assess the overall quality of the analyses. Also, the supplementary materials describe the QC and processing details but details are missing regarding the statistical analyses of these tumor tissue studies.

We relocated the key methodological details into the main text and refined the description of the processing details to address these issues. The Materials and methods section now includes implementation details of the UKBB sample and genotype QC, criteria use-d to identify UL cases, and the model used for association testing. Further key details on the meta-analysis, genomic risk score (GRS) and follow-up analyses were also relocated from supplementary material to the main text.

Many of their functional analyses (e.g., gene expression, telomere length, and DNA methylation) involve comparing uterine fibroid samples to 'control' samples that are not well-defined. The authors should indicate whether the control tissue is patient-matched unaffected, adjacent myometrium or myometrial tissue isolated from a completely unaffected control.

We have clarified the text regarding the sample types involved in the follow-up analysis. All the controls were patient-matched, unaffected myometrium. Please see the Materials and methods section for exact numbers of each sample type.

In the discussion of gene expression analyses, the authors mention accessing 'myometrium' tissue on the GTEx Portal (Discussion, fourth paragraph). Currently, GTEx only provides data taken from the entire uterus. Moreover, given the nature of the individuals from whom GTEx tissue is derived (i.e., predominantly older individuals dying of natural causes), the relevance of these analyses to uterine gene expression in reproductive-aged women is questionable.

We understand this point. To be on the safe side, these references and the respective parts of discussion were removed from the Discussion section.

3) Statistical analysis:In general the “Statistical analysis” Materials and methods section is not presented with sufficient detail in the main text.

We relocated the most relevant sections of the supplementary methods into the main text. This included all the statistical methods used for GWAS, meta-analysis and GRS association, as well as details on the eQTL and meQTL identification.

Why did the author's perform Bonferroni corrections of the GWAS but FDR on other analyses? Would those analyses survive a Bonferroni Multiple testing correction?

FDR was chosen whenever the total number of tests was expected to be large, that is, a family of one hundred or more multiple hypotheses. Controlling for FWER at this scale would lack discovery power.

GWAS meta-analyses across all three stages should be presented including heterogeneity statistics.

We added supplementary files with the requested information: for each cohort, we report the allele frequency, association effect size and P-value of each of the 57 GRS SNPs. Supplementary file 1 gives these statistics together with heterogeneity estimates, Cochrane's Q and I^2^ index, for the UKBB and Helsinki cohorts. Supplementary file 2 gives the SNP statistics for UKBB, Helsinki and NFBC. Supplementary file 3 contains these information for the five UKBB follow-up cohorts.

4) Ancestry and risk:The authors use self-reported ancestry in their follow-up analyses. PCA-based analysis of ancestry-informative markers is necessary for ensuring homogeneity of the individual follow-up cohorts. The discovery and stage 2 meta-analyses were performed primarily in white European populations and did not replicate in the final stage 3 that included multiple ancestries, including those of African ancestry. The author's main conclusions are that their findings explain the occurrence of leiomyoma in women of African origin (both in the abstract and final discussion). Correlation between increased percentage of African ancestry and probability of reporting uterine fibroids is a necessary analysis for the conclusion that "[Their] findings… in part explain the more frequent occurrence of UL in women of African origin."

The requested PCA-based analysis is now included: a short summary was appended to the Materials and methods section, together with new supplementary material. In short, we combined 21 panels of ancestry informative markers – in total 1,396 unique SNPs – and compared the resulting PCs against the self-reported ancestry. As expected, the genetic differences at these informative markers separated each self-reported ancestry into a distinct, homogeneous cluster. Of note, the first principal component differentiated both the ‘African’ and ‘Black Caribbean’ cohorts from the others. Please see Appendix—Figure 4 for the resulting clustering.

To address reviewer’s concern on “correlation between increased percentage of African ancestry and probability of reporting ULs”, we note the following: First, the observed proportions of reported UL cases in the ‘African’ and ‘White British’ cohorts (19.1% and 7.0%, respectively) do match with the literature estimate of a 2-3 times higher prevalence of ULs in African compared to Caucasian ancestry (see also the references in the Results section).

Second, a correlation between increased percentage of African ancestry and reported ULs has seen overwhelming evidence in literature, and to no surprise, we see a strong support for it also in our data. We extended the above ancestry informative marker analysis: the first principal component was chosen as a proxy for the degree of African ancestry, and a comparison of the 2,212 UL cases and 21,054 controls did show a significant difference in their distributions, that is, the prevalence increases in degree of African ancestry (P<2.2⨉10^-16^; Wilcoxon rank sum W=3⨉10^7^) as expected. See Appendix—Table 1 for the background information on each follow-up cohort.

The authors should describe the frequencies of the variants evaluated in stage 2 and 3 for their respective populations. Is it possible that some of the variants that associated in stage 1 in European ancestry individuals are rare in other ancestral groups?

We added supplementary files with the requested information. The follow-up cohorts do carry some differences regarding their allele frequency and heterogeneity of effect size as expected (Supplementary file 1-3). To answer reviewer’s concern, we note that the mean GRS was relatively even across the different population-strata among both the follow-up cohorts (Figure 4) and gnomAD (Appendix—Figure 6). That is, none of the follow-up populations displayed a deflated GRS or exceptionally rare risk alleles.

The interchanging usage of 'African American', 'Admixed American', and 'African' in the text should be addressed.

We unified the terminology as requested.

5) Correlations of findings with biology:Recent work investigating the role of the WNT4/CDC42 locus in endometriosis has shown that CDC42 – not WNT4 – contributes to the endometriosis phenotype. In light of the evidence drawn from analyses in endometriosis, the work presented in this manuscript provides insufficient evidence to implicate WNT4 in uterine fibroids. Additional work to either verify or strengthen the specific implication of WNT4 (from the CDC42/WNT4 locus) is necessary, else the revised manuscript should more fully justify the conclusions around WNT4.

*WNT4* (and WNT/β-catenin signaling pathway) plays a significant role in the myomagenesis of *MED12* mutated leiomyomas. In addition, the evidence in our tumor data – risk SNP associating with increased *WNT4* expression but not *CDC42* expression – leans towards *WNT4* as the predisposing gene. However we do agree that this evidence is not unambiguous and have revised the discussion to better acknowledge the possible role of *CDC42* (see also Table 1 and Figure 2).

A more thorough and solid functional validation to prove the conclusions should be performed. Given the high frequency of MED12 mutations in leiomyomas the heritability to MED12 mutations is possibly one of the strongest points of the manuscript, however no real validation has been demonstrated in the study. The authors show a "trend" of an over-expression which cannot be taken as enough evidence for such a claim. What was the effect size in patients with both MED12 mutated positive and negative leiomyoma? A strong association between the risk allele rs5937008 and the MED12 status is described and the authors argue that the risk allele could facilitate selection of somatic MED12 mutations, possibly through increased expression.

We recognize that the expression trend is not enough evidence for the claims and thus such wording is removed from the discussion. The effect size with regard to expression was larger in wild type tumors compared to *MED12* mutated tumors (0.03 and 0.003 respectively). Thus if the expression difference is of significance it would seem that the risk SNP confers higher *MED12* expression, but after the mutation this addition loses some of its selective power (as the impact of expression level is secondary as compared to the effect of the mutation). This is of course very speculative and indeed the relation between *MED12* expression and rs5937008 is not discussed in the revised paper.

UL can be differentiated in 3 mutually exclusive subtypes, with the MED12 mutant subtype been the most common. In the manuscript the authors refer and analyze carefully associations between risk alleles and MED12 mutation positive samples, however there is no mention to the other two subtypes. From the set of 1481 lesions included in the study, what was the distribution of tumors and patients with FH deficient or HMGA2 overexpressing lesions? Were FH mutation carriers included in the study? Was there a risk allele from the 22 significant loci that associated to any of these subtypes?

Other subtype-specific associations were not evaluated due to the GRS association merely to patients with *MED12* mutated lesions. Efforts to scrutinize the more rare subtypes will also require larger numbers of these subtype cases as discussed in the manuscript.

Although I understand that not all associations can be experimentally proven, a functional evaluation of this phenomenon and how would the rs5937008 drive MED12 mutations would greatly improve the manuscript and provide strong evidence of the rs5937008 allele and UL development. These data would be of great importance towards clinical management of patients.In the same fashion that for rs5937008, it would be of great interest to confirm a functional effect by which rs58415480 regulates ESR1 expression or the repercussion of the lead SNP at 2p on GREB1 mediated regulation of ESR1.

These are truly important aspects but, unfortunately, not feasible in the scope of this manuscript. Typically, taking GWAS findings to this level has taken years of work.

TERT SNPs rs72709458; rs2736100; rs2853676 are known to be related to many different neoplasias. How did those relate to MED12 mutated tumors?

We made no a priori hypothesis regarding these specific SNPs and *MED12* mutated tumors. That said, when controlling for multiple testing over all 57 GRS SNPs, the above *TERT* SNPs did not show significant associations to the *MED12* status. See Appendix—Table 5 for the details and any nominal associations.

The authors divide the 22 risk loci into two global biological processes one of which is genomic instability and telomere maintenance. Genetic instability is a common mechanism driving tumorigenesis and molecular signatures of genomic instability can be extrapolated from tumor profiling, in this context, a molecular profile of the leiomyomas associated with ATM, TP53, TERT, TERC or OBFC1 could reveal strong evidence to validate the role of genetic instability in UL development.

As mentioned in the manuscript – and further clarified in the revised version – we were not able to see a correlation between genetic instability and the risk SNPs. This could be due to the lack of power in whole-genome analysis as well as the technical limitations (e.g. inability to detect short copy number changes and copy neutral translocations) of SNP array data.

6) Additional issues in interpretation:The follow-up analyses rely solely on the composite GRS as a surrogate measurement for genetic association with the phenotype of interest. It is assumed that this limited metric is selected due to the underpowered nature of the small follow-up cohorts.

The composite GRS approach has seen a wide range of applications in genetics research and, indeed, the approach was chosen due to the limitations of the follow-up cohorts.

None of the results presented in the 'Association to Gene Expression' section are actually significant. The authors should consider removing this section from the main portion of the manuscript. Similarly, the 'negative trend' observed in the 'combined effect of TERT, TERC, and OBFC1' on telomere length is also insignificant. The fact that the combined effect of three major genes still is insignificant indicates some concern with interpretation of the results.

We thank the reviewers for these considerations. We choose to retain both the eQTL and the telomere length results. In the eQTL analysis, our discovery power is limited due to the combination of a large family of hypotheses and restricted number of samples per genotype. While restricted number of samples is also a problem in the telomere analysis, some of the associations do reach statistical significance (e.g. the effect of rs2736100 on telomere length). All in all, we trust that the reported data does add value towards designing future and more focused experiments.

The authors state that they replicated in NFBC and in the other ethnic groups but the replication results for the other five ethnic groups were not in the supplementary material. Those should be provided. In addition, the authors state (subsection “Replication of the GWAS and GRS”) that the third stage replicated the results but later on in the same paragraph state that no associations passed multiple testing corrections for the single SNP associations. The authors should state clearly that one of the associations replicated in stage 3, based on the evidence they present.

The supplementary material did already include all the GRS-based analyses and summary statistics in Appendix—Table 6*(“Summary of all the association tests for GRS”*) and Appendix—Figure 5 (“Association between GRS and phenotype”). These show the case-control distribution of GRS in each of the six follow-up cohorts and the resulting association statistics. The GRS association to the UL phenotype did replicate in NFBC and in the five other ethnic cohorts.

Regarding the single SNP association tests, we followed the reviewer’s request and extended the supplementary material to include SNP summary statistics for all the cohorts. We have clarified the statement regarding single SNP associations and multiple testing correction (subsection “Replication of the GWAS and GRS”, second paragraph). Due to the limited size of the follow-up cohorts, a single SNP was underpowered to capture the phenotype association, thus, the above mentioned composite GRS was utilized to model UL risk and replicate the phenotype association. That said, we have now clarified the main text to make a more clear separation between the single-SNP and GRS based results and their interpretation.

There are multiple locations where the authors state that results significantly associated but later on state that the results did not pass multiple testing corrections. The authors should make it clear that the there was no evidence for significant association if effects did not pass multiple testing corrections. This language is also present in the Discussion, where results are described as replicating in stage 3.

We have improved the presentation and made the interpretation of results easier to follow. The reviewer’s concern was addressed by making a more clear distinction between single SNP and GRS based results and their interpretation.